# Export competitiveness network of Chinese provincial agricultural products: E volution, performance, and influencing factors

**Ye Chen** [ID]*, **Zhuoxi Liu**[©], **Zhongming Yin**[©]

School of Economics, Southwest Minzu University, Chengdu, Sichuan Province, P. R. China

[©] These authors contributed equally to this work.
* cherryheipi@163.com

**Data Availability Statement:** All relevant data are within the manuscript and its Supporting Information files.

**Funding:** This research was funded by two grants: the Social Science Planning Project of Sichuan

## Abstract

The development of top-tier agricultural product bases and demonstration zones aims to propel Chinese agriculture into the global market, form international competitiveness, and advance to the high end of the global industrial and value chain. This study contributes to the existing literature by investigating the performance and factors influencing the competitiveness of Chinese agricultural product exports at the provincial level, using a complex network perspective, and offers recommendations for improvement. In this regard, the article uses agricultural product export data from 2015 to 2022 to create export competitiveness networks of Chinese provincial agricultural products. Our analysis revealed four primary observations. Firstly, significant opportunities exist to improve the competitive edge of agricultural product exports across various Chinese provinces. Secondly, with the exception of Guangdong Province, the types of competitively robust agricultural products exported by each province are inconsistent. Thirdly, international sister-city relationships can foster the development of an export competitiveness network of Chinese provincial agricultural products, while geographic distance may hinder this network. These findings are consistent. Lastly, the development of this competition-focused export network is also influenced by province-level and national characteristics, and these influences have unique differences. All provinces should strive to produce distinctive and globally competitive agricultural goods to build their competitive advantage in international trade. They should also implement strategic regional planning to support the global expansion of their agricultural products.

## Introduction

As China enters a modern phase of international agricultural cooperation, it has made significant strides in cultivating mutually beneficial relationships and promoting the Belt and Road Initiative. According to the Food and Agriculture Organization of the United Nations, in 2022, China emerged as the world's largest importer and the sixth-largest exporter of agricultural products, amassing agricultural imports amounting to $259.1 billion and exports totaling $80.3 billion.

Province, titled "Research on the Comparative Advantage Structure Evolution and Development Strategies of Sichuan Province's Agricultural Products Export under the Belt and Road Initiative" (Grant No. SC22C012), and the Fundamental Research Funds for the Central Universities at Southwest Minzu University, for the project "Study on the Evolution of China's Foreign Trade Pattern of Crop Seeds and Risk Prevention Strategies" (Grant No. 2022SJQ004).

**Competing interests:** The authors have declared that no competing interests exist.

In 2022, China's Ministry of Agriculture and Rural Affairs rolled out the "14th Five-Year Plan for International Cooperation in Agriculture and Rural Affairs". This plan emphasized the importance of enhancing international collaboration and fostering competitive advantages in agricultural trade. Focusing on competitive advantage has become a significant factor in China's agricultural cooperation endeavors [1].

Despite this progress, Chinese agriculture contends with challenges such as limited resources, land fragmentation, and high production costs [2]. These issues have led to a comparative disadvantage in Chinese agricultural exports [3,4] and hindered export growth and sustainable development [5]. Notably, the primary competitive agricultural products in the global market include wheat from Russia, soybeans from the United States, and rice from Thailand. As a major agricultural nation, China is relatively deficient in globally competitive agricultural products.

The competitive advantage of export products is crucial for the high-quality economic development of a country or region. Firstly, it stabilizes export relationships and is influential in subsequent expansion [6]. Secondly, it enhances competitiveness in the global market and improves the international economic circulation system [7]. Due to variations in geographic location and resource endowment, Chinese exported agricultural products exhibit distinct geographical characteristics and imbalances. Therefore, it is essential to provide complete insights into the competitive advantages of provincial agricultural exports [8]. Moreover, to identify the factors that propel the effective international expansion of these advantageous agricultural products [9,10]. This identification will help establish new competitive edges for provincial agricultural products and foster their entry into global trade. Further research can then probe into merging the export of provincial agricultural products with the high-quality development of Chinese agriculture [11].

The main objective of this study is to investigate the evolution, performance, and influencing factors of the export competitiveness network of Chinese provincial agricultural products. In this context, provinces refer to 27 Chinese provinces and 4 municipalities directly governed by the Central Government. Exploring the evolution of the export competitiveness network of agricultural products and identifying the influencing factors will help promote the high-quality development of agricultural trade in Chinese provinces. However, the existing literature lacks a focused and detailed exploration of the export competitiveness of agricultural products from Chinese provinces. This study fills this gap by conducting a comprehensive analysis from both micro and macro perspectives.

The innovations presented in this study are primarily seen in several aspects. (1) Most existing literature studies the export competitiveness of Chinese agricultural products from a national perspective. This paper, however, examines the international competitiveness of Chinese agricultural products at the provincial level, highlighting the differentiation of competitive agricultural products across Chinese provinces. (2) By applying a two-mode network perspective and an enhanced revealed comparative advantage index, this paper innovatively constructs an export competitiveness network of provincial agricultural products. This approach more effectively identifies competitive agricultural products across provinces and further enriches the research paradigm concerning trade relationships. (3) Using the Exponential Random Graph Model (ERGM), this paper investigates the factors influencing the development of the export competitiveness network from a comprehensive perspective. Compared to the conventional regression models, ERGM is better suited to track the endogenous dependencies in forming the export competitiveness network.

The remainder of this paper is structured as follows: Section 2 reviews the existing literature and highlights the contributions of this study. Section 3 describes the establishment of an export competitiveness network and the structural indicators for provincial agricultural

products. Section 4 explores the characteristics and evolution of the network structure. Section 5 presents our empirical analysis and discussion. Section 6 concludes the paper with our main findings and policy recommendations.

## Literature review

Few articles have systematically analyzed the competitive strengths and enhancement strategies of agricultural exports across Chinese provinces, explicitly considering the classification of agricultural products. The majority of research in this field still emphasizes the general situation of Chinese agriculture.

### Related research on the competitive advantage of Chinese agricultural exports

China has developed competitive advantages in industries that rely on low labor costs, resource availability, and environmental factors since the implementation of economic reforms and opening-up policies. This has resulted in labor-and resource-intensive sectors, such as textiles and clothing, becoming the most competitive industries [12]. Additionally, as a major agricultural nation, China's trade in agricultural products significantly influences its foreign trade.

However, the share of agricultural products in China's Gross Domestic Product (GDP) has declined in recent years, weakening its international competitiveness for these products [13]. The Association of Southeast Asian Nations (ASEAN) possesses a stronger revealed comparative advantage (RCA) of agricultural product than China, showing that Chinese agricultural products fall short in international competitiveness [14].

China's accession to the World Trade Organization (WTO) and the resulting trade liberalization have notably affected its agricultural trade balance, causing a trade deficit and reduced competitiveness in specific product categories [3]. Furthermore, rising production costs and a diminishing demographic dividend, which China once benefited from, undermine its competitive edge in exporting agricultural products.

The advancement of technology is crucial for maintaining a competitive edge [15]. Traditional Chinese agricultural products, such as tea and live pigs, remain competitive in the international market [5]. This suggests that China needs to increase its investment and intensify efforts to develop competitive agricultural products.

### Research on the network of agricultural product export

More countries and regions are joining the agricultural trade network [16]. This network plays a significant role in global trade, with network analysis methods revealing the structure and patterns [17]. The network structure, defined by elements like centrality and modularity, can indicate the evolution of communities and their leading countries [18]. The density of the global grain trade network keeps increasing, driven by the nodes of grain export [19]. Countries with high betweenness centrality serve as hubs in agricultural trade networks and tend to establish a greater number of trade relationships [20]. This phenomenon results in more network star effects [21]. Furthermore, the star effect promotes the establishment of additional network relationships [22]. In addition, attribute variables such as factor endowment, economic size, and geographic location are critical factors influencing the expansion of agricultural trade networks [23].

In the agricultural products export network of Belt and Road, there is more complementarity than competition between countries [24]. China has become the leading importer in both physical and value trade networks and the principal exporter in the value network [25]. However, there is a growing reliance on trade with various countries for some agricultural products,

such as soybeans. This calls for enhancing self-sufficiency in soybean production to reduce reliance on imports [26].

## Research on the two-mode network

Initial research into two-mode networks emphasized clustering, density, and centrality metrics [27]. Since then, indicators of structural features and empirical econometric models have been developed [21,28]. Researchers have advanced the study of two-mode networks from a static to a dynamic framework [29]. Various practical applications of two-mode networks, such as firm-city networks [30,31], debt-bonding networks between non-financial and financial listed companies [32], networks mapping the participation of Southern women in activities [33], stakeholder and cost management networks [8], and fund-stock networks [34], have been thoroughly explored. These studies typically involve the conversion of a two-mode network into two one-mode networks, assessing the structure of the two-mode network, and analyzing the influencing factors.

In summary, while the current literature provides extensive theoretical and empirical research on the competitive advantages of agricultural trade and trade networks, it lacks a focused and detailed examination of the export competitiveness of agricultural products from Chinese provinces. Although some studies have explored the export competitiveness of Chinese agricultural products [5], none have assessed the influencing factors and evolution of competitiveness from a network perspective. This gap hinders our comprehensive understanding of the export networks of provincial agricultural products. As a result, this paper struggles to analyze the strengths and weaknesses of provincial exports and to develop high-quality strategies for agricultural product exports across different provinces. With these considerations in mind, this article aims to study the high-quality development of agricultural trade in China through the lens of the "dual circulation" of international and domestic markets. Based on provincial export data, this article aims to establish a competitive network for agricultural product exports to various countries and regions. The objective is to analyze the evolution, performance, and influencing factors of the export competitiveness network of Chinese provincial agricultural products.

This article extends the existing literature by contributing to two vital areas. Academically, it addressed the evolution of the export competitiveness network of agricultural products in Chinese provinces and identified its driving factors from a two-mode network perspective. The study adopts a more nuanced approach to examine the high-quality growth of China's agriculture, offering innovative research methodologies and perspectives. Practically, it substantiates policy-making around the export of agricultural products from various provinces in China. Furthermore, it aids their assimilation into the high-quality development of the Belt and Road Initiative, providing an advanced scientific foundation.

## Theoretical analysis and hypothesis

### Network pure structural effect

Network relationships characterized by inherent interdependencies foster the development of additional relationships, ultimately culminating in a comprehensive network. This phenomenon is called the "network pure structural effect" [35,36]. Pure structural metrics, such as edges, 2-stars, 3-stars, 3-paths, and 4-loops, effectively conform to the Markov assumption [21]. The star structure is regarded as a "convergence" effect or a "rich get richer" effect [37], which increases the probability of connected entities establishing more distant connections [22]. In other words, nodes with high network neutrality are more likely to form additional connections. For Chinese provinces, this implies they are inclined to form new relationships

with other countries and regions after establishing competitive export relationships for agricultural products. Based on these observations, this paper proposes the following hypothesis.

H1: The star effect of the export competitiveness network of provincial agricultural products contributes to the formation of network relationships.

## Node attribute effect

Individual quality plays an important role in forming network relationships [38]. The principle of homophily suggests that individuals are more likely to establish connections with those who share similar attributes, thereby influencing the probability of network connectivity [39,40]. Consequently, this principle substantiates the impact of individual attributes on forming network relationships within social networks.

Economic attributes significantly influence trade and the overall structure of trade networks [41]. For instance, a country's level of openness and reciprocity can affect the establishment of wheat trade relations [23]. Likewise, the intensity of research and development in technology, along with a province's technological openness, are key determinants in the evolution of interprovincial trade networks [42]. From these observations, the following hypothesis is formed.

H2: Provinces and economies with larger populations and higher levels of economic development are more inclined to establish competitive export trade relationships for agricultural products.

## External covariate effect

Sociologists argue that an individual's social capital comprises the resources available within their social relationships [43], including the sum of existing and potential resources in their social network [44]. This social capital can foster collaboration and provide beneficial resources to an individual [45]. It is fundamentally defined by the network of social relationships [46] and can be considered an investment in such relationships [47]. Economically, social capital is understood as a broader trust [48]. Individuals can leverage social capital relationships to enhance their advantages and seize opportunities [49].

In the realm of local governmental diplomacy, international sister-city relationships enable cities to accumulate social capital in their host country. Sister-city relationships are an advantageous resource [50]. By enhancing knowledge of foreign markets [51], international sister-city relationships assist in reinforcing identity and trust. The quality of information and resources transmitted by such strong relationships generated by identification and trust tends to be higher, providing both parties with a competitive advantage in trade and economic activities [43]. Furthermore, it reduces trade disputes and friction, thereby influencing the establishment and robustness of trade relationships [52]. As a result, products from friendly cities or provinces have a significant advantage in entering the markets of friendly countries.

Geographic location influences the development of relationships [22]. Usually, distances are used to proxy costs at an abstract level [53]. As a result, coastal cities possess a greater geographical advantage when exporting their products [54]. Longer bilateral geographical distances increase trade costs, thereby becoming an impediment to product exports [55] and further hindering the establishment of competitively advantageous trade relationships. Based on these considerations, this study puts forth the following hypotheses:

H3: International sister-city can contribute to the formation of relationships in the export competitiveness network of Chinese provincial agricultural products.

H4: Bilateral geographical distance may be an impediment to the establishment of relationships in the export competitiveness network of Chinese provincial agricultural products.

## Research design

The article utilizes data from Chinese Customs to examine the dynamic evolution and competitive edge of agricultural exports across Chinese provinces. This data has been available since 2015. By 2023, it may be possible to obtain complete data for the year ending in 2022. Therefore, the research sample in this paper is based on data from 2015 to 2022 to ensure consistency and coherence.

Complex network analysis are leveraged to build competitiveness networks of provincial agricultural exports and evaluate network structure. Complex networks, an emerging field in complexity research, explore the properties of intricate systems by abstracting them into networks of interconnected nodes and edges [56]. This approach adopts topological indicators like loop numbers, node strength, clustering coefficient, and betweenness centrality to highlight node heterogeneity, structural complexity, and dynamic variability [57]. Therefore, employing complex network analysis can effectively illuminate the structural evolution of the export competitiveness network and identify the competitive agricultural products. In contrast to traditional research methods, complex network analysis can investigate the key factors influencing agricultural export competitiveness at various levels. Based on this, we can provide actionable recommendations to enhance the export competitiveness of these products in Chinese provinces.

## Network construction

This paper builds a two-mode network of Chinese provincial agricultural exports. The nodes in the two-mode network comprise two types. The first type are composite nodes encompassing provinces and agricultural products. Each node in first type corresponds to a province and the agricultural product it exports. For example, if Guangdong Province exports vegetable oil, the node would be represented as Guangdong Vegetable Oil. This paper identifies 189 agricultural product categories classified at the HS 4-digit level. The advantage of classifying agricultural products at the HS 4-digit level is that it effectively distinguishes between various agricultural products without leading to excessive granularity. In this paper, we have excluded the Hong Kong, Macao, and Taiwan regions of China. Therefore, the theoretical maximum number of these nodes is 189 multiplied by 31. The second node type represents the export destination, signifying a country or region, denoted simply as the country in this paper. We excluded the sample of countries with missing data on economic freedom for more than two consecutive years, resulting in 186 countries. From this group, we further excluded countries that had missing GDP data for more than two consecutive years. Ultimately, the final nodes comprised 174 countries.

The edges of the two-mode network of Chinese provincial agricultural product exports are the trade relationships established between the provincial agricultural products and various countries. See Fig 1 for a diagram depicting the construction of this two-mode network for provincial agricultural exports.

Based on the construction of a two-mode network for provincial agricultural exports, this paper develops the Trade Competitive Advantage (TCA) index by integrating the RCA index introduced by Balassa [58] with the methodology proposed by Feng et al. [59]. The TCA index assesses the competitiveness of provincial agricultural products in international markets.

Feng et al. [59] primarily researched RCA networks along the Belt and Road without categorizing the products, which are classified as one-mode networks. In contrast, this paper examines the export competitiveness network of different agricultural products from a provincial perspective, categorizing it as a two-mode network. This micro two-mode network is particularly suitable for analyzing the export competitiveness of Chinese provincial agricultural

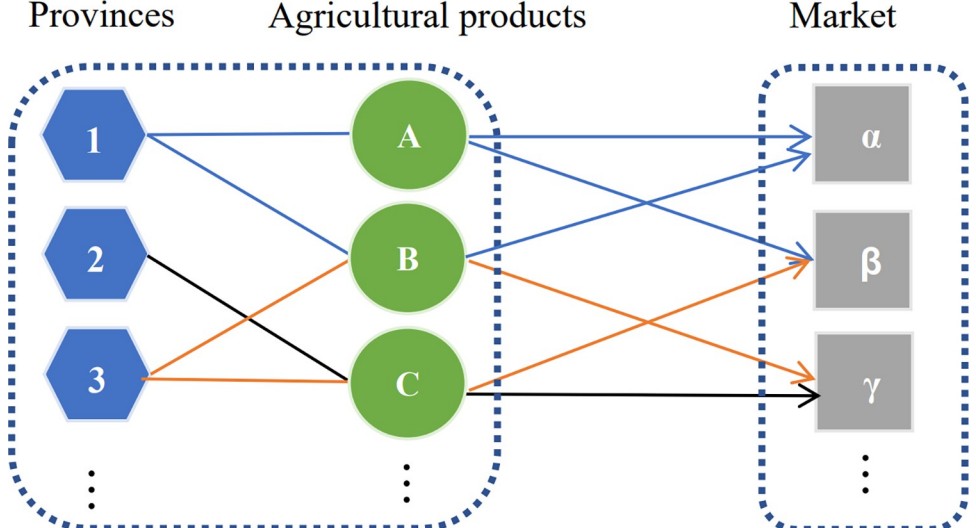

**Fig 1. A schematic diagram of the construction of a two-mode network for provincial agricultural exports.**

products. Based on the trade volume of various agricultural products exported from Chinese provinces to countries, the TCA is constructed as follows:

$$TCA_{ijk}^t = \frac{w_{ijk}^t}{\sum_{k(j)} w_{ijk}^t} \Big/ \frac{\sum_{i(k)} w_{ijk}^t}{\sum_i \sum_j w_{ijk}^t} \tag{1}$$

In the formula, $w_{ijk}^t$ represents the quantity of agricultural products $j$ exported from province $i$ to country $k$ in year $t$. $\sum_{k(j)} w_{ijk}^t$ represents the total export value of agricultural product $j$ shipped from province $i$ to all countries in year $t$. $\sum_{i(k)} w_{ijk}^t$ represents the total export volume of agricultural products $j$ from all provinces in China to country $k$ in year $t$, and $\sum_i \sum_j w_{ijk}^t$ represents the total amount of agricultural products exported by all provinces in China to country $k$ in year $t$. The value range of $TCA_{ijk}^t$ is $[0, +\infty)$. The larger the value, the more competitive the agricultural product $j$ of province $i$ has in the country $k$ compared to other provinces in year $t$. Take a threshold value of 1.25 for the index, which means that values with $TCA_{ijk}^t$ greater than 1.25 are retained as edges of the export competitiveness network $G(V, TCA)_t$. $G(V, TCA)_t$ is an $M*N$ matrix, where $M$ represents the composite nodes of provinces and the agricultural products they export, and $N$ denotes the nodes of destination countries. The edges of the export competitiveness network are the $TCA_{ijk}^t$ index with a value greater than 1.25. A $TCA_{ijk}^t$ index greater than 1.25 does not prioritize a province with a high export of agricultural products as having a competitive advantage in destinations. Instead, it focuses on the share of specific agricultural exports in the province and the share of national imports of specific agricultural products in total imports. Therefore, the $G(V, TCA)_t$ can precisely define the agricultural products for which each province has maintained a competitive advantage in destinations over the years. Based on this, this paper can elaborate on the structural evolution of the export competitiveness network of provincial agricultural products.

## Construction of structural indicators

**Number of network nodes, edges, density, and average degree.** A two-mode network consists of two types of nodes that collectively determine the network's size. In the export

competitiveness network of provincial agricultural products, the nodes include the number of provincial agricultural products ($M$) and the number of countries ($N$). The total number of network nodes is the sum of these two types of nodes and can be calculated using the following formula:

$$Nodes = M + N \tag{2}$$

The edges ($E$) in a two-mode network represent the connections between two distinct types of nodes. Specifically, the number of export relationships with competitiveness between provincial agricultural products and countries equals the sum of the degrees of provincial agricultural products ($\sum_{ij=1}^{M} d_{ij}$) or the sum of the country degrees ($\sum_{k=1}^{N} d_k$). The formula is calculated as follows:

$$E = \sum_{ij=1}^{M} d_{ij} = \sum_{k=1}^{N} d_k \tag{3}$$

The density of a two-mode network is defined as the proportion of actual edges $E$ to the maximum possible edges ($M*N$) in the network. This density mirrors the proximity between the two types of nodes within the network. The closer the index is to 1, the more connected the network becomes. In the export competitiveness networks of provincial agricultural products, network density quantifies the ratio of the actual number of dominant export relationships to the maximum potential number. Here is the calculation formula:

$$Density = E/(M*N) \tag{4}$$

The average degree ($LPN$) of a two-mode network is determined by the ratio of the number of edges to the number of nodes. This metric is also applicable to the average degrees of various node types and helps characterize the network's sparsity. In the export competitiveness network of provincial agricultural products, the average degree pertains to the average amount of connections each node has made with different node types. The calculation formula is as follows:

$$LPN = E/(M + N) \tag{5}$$

Generally, the size of a two-mode network is determined by the number of nodes and edges. As the number of these metrics increases, so does the size of the network. Thus, the network's trend can be evaluated based on these metrics. The network's density and average degree provide valuable insights into the strength of connections within an export competitiveness network of provincial agricultural products.

**Degree of nodes.**   In two-mode networks, the node degree is defined as the number of relationships established between a node and all nodes of a different type. It is commonly utilized to measure the centrality of a node within the network. A higher node degree indicates greater social status and power [60]. In the export competitiveness network of provincial agricultural products, the node degree represents the number of competitive export relationships between the provincial agricultural product and the country. In the network, $d_{ij}^t$ is the number of competitive export relationships between provincial agricultural product $ij$ and countries in year $t$, which is the degree of provincial agricultural product $ij$ in year $t$. $d_k^t$ is the number of competitive agricultural products exported from different provinces to country $k$ in year $t$, which is the degree of country $k$ in year $t$. By calculating the node degrees, we can further elucidate the competitive advantages of agricultural products exported by various provinces in China to the international market.

## Network structure characteristics and evolution analysis

The agricultural products exported from various provinces have proven competitive within expanding international markets, primarily evidenced by increased network edges and nodes. This suggests an overall improvement in the competitiveness of agricultural products.

On one hand, most of the 174 countries analyzed in this article participated in the network, peaking at 158. This suggests that agricultural products exported by Chinese provinces can develop competitive advantages in these countries, exhibiting expansive coverage. On the other hand, the number of provincial agricultural products included in the network has consistently remained high, making up over 99% of all provincial agricultural products. This indicates that nearly every agricultural product exported by Chinese provinces possesses a competitive advantage in the international market. See Table 1.

There is a notable potential for improving the competitive advantage of agricultural exports from Chinese provinces. While international trade networks usually exhibit high density [61], the density of the export competitiveness network this study constructed is remarkably low, consistently below 0.039. Moreover, a consistent uptrend in $LPN$, $LPN(M)$ and $LPN(N)$ indicates an increase in competitive agricultural export partnerships. On average, each provincial agricultural product shows a competitive edge in merely five or six countries. The provinces selectively choose specific export markets for their agricultural products to cultivate a competitive advantage. Thus, the competitiveness of agricultural product exports from Chinese provinces is essentially minimal. A massive push is required to enhance the export of high-quality, competitive agricultural products from Chinese provinces. The main provinces that have established many competitive agricultural export relationships are Shandong, Guangdong, Jiangsu, Zhejiang, and Fujian. See Fig 2.

Table 2 presents the top 5 node degrees for countries and provincial agricultural products.

The numbers in parentheses represent the node degree. A higher node degree for a country indicates a greater number of competitive agricultural products available in the market. Similarly, a higher node degree for a provincial agricultural product suggests a stronger competitive position in export markets. The top 5 countries in node degree perform consistently. Leading this list are Japan, the United States, South Korea, Malaysia, Thailand, and Vietnam. Japan and the United States have invariably held the first and second positions, indicating their pivotal role in the network.

**Table 1. Network structure of the export competitiveness network of provincial agricultural products from 2015 to 2022.**

| Year | E | Density | LPN | Countries level | | Provincial agricultural products level | |
|------|------|---------|------|------|--------|------|--------|
| | | | | M | LPN(M) | N | LPN(N) |
| 2015 | 13704 | 0.035 | 4.72 | 140 | 97.89 | 2764 | 4.96 |
| 2016 | 14286 | 0.034 | 4.83 | 146 | 97.85 | 2809 | 5.09 |
| 2017 | 15257 | 0.036 | 5.10 | 149 | 102.40 | 2840 | 5.37 |
| 2018 | 15484 | 0.036 | 5.20 | 152 | 101.87 | 2825 | 5.48 |
| 2019 | 16406 | 0.036 | 5.39 | 158 | 103.84 | 2887 | 5.68 |
| 2020 | 16169 | 0.039 | 5.43 | 148 | 109.25 | 2832 | 5.71 |
| 2021 | 16540 | 0.038 | 5.46 | 151 | 109.54 | 2881 | 5.74 |
| 2022 | 17396 | 0.037 | 5.50 | 157 | 110.80 | 3004 | 5.79 |

Data source: Calculated by the author based on Chinese Customs data.

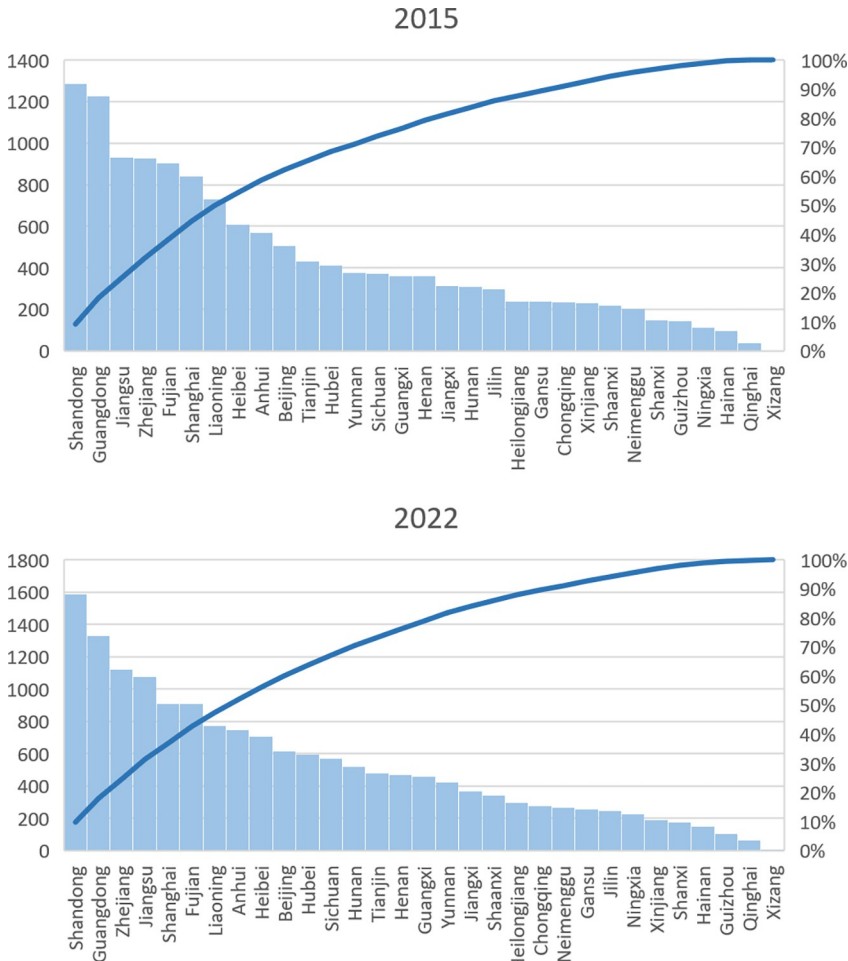

**Fig 2. Pareto chart of the number of competitive agricultural export relationships established in each province in 2015 and 2022.** (data source: Chinese Customs data).

This suggests strong competition for agricultural products exported from Chinese provinces to Japan and the USA. Typically, Japan relies heavily on imports of agricultural products, with China emerging as a significant source. The prospect of Chinese agricultural products forming a competitive edge in the Japanese market is high. In 2022, for instance, a total of 1,131 Chinese agricultural products were identified as competitive in Japan. According to data from the U.S. Department of Agriculture's Foreign Agricultural Service, in 2022, China was the third-largest foreign supplier of agricultural products to the United States, following Canada and Mexico. Some products like Qinghai Cheese, Jiangsu Natural Forage, and Guangxi Coconut exhibit a formidable competitive advantage in the U.S. market.

Moreover, China has forged lasting agricultural trade alliances with countries such as South Korea and Malaysia, solidifying its position as one of the top three trading nations for both importing and exporting agricultural products.

The top five provincial agricultural products, ranked by node degree over the years, are mainly from Guangdong and include Vinegar and Vinegar Products, Margarine, and Vegetable Oil. These products suggest a more decisive competitive edge in the international market. Guangdong, a major agricultural production and export hub, has actively promoted its inimitable agricultural products globally. This strategic move has significantly escalated the overall

**Table 2. Top 5 countries and provincial agricultural products in degrees ranking of export competitiveness network from 2015 to 2022.**

| Year | Top 5 countries in degree ranking | Top 5 provincial agricultural products in degree ranking |
|---|---|---|
| 2015 | Japan (1084), United States (947), South Korea (870), Malaysia (767), Thailand (603) | Guangdong Vinegar and Vinegar Products (58), Guangdong Vegetable Oil (42), Guangdong Cocoa Food (39), Shandong Beer (35), Guangdong Anise (33), Guangdong Roasted Cake (33), Hebei Plant Branches and Leaves (33) |
| 2016 | Japan (1090), United States (1015), South Korea (898), Malaysia (790), Thailand (651) | Guangdong Vinegar and Vinegar Products (58), Guangdong Vegetable Oil (45), Guangdong Cocoa Food (36), Guangdong Other Tobacco (35), Jiangsu Lanolin (35) |
| 2017 | Japan (1082), United States (1018), South Korea (865), Malaysia (767), Thailand (669) | Guangdong Vinegar and Vinegar Products (60), Guangdong Vegetable Oil (44), Jiangsu Cocoa Powder (41), Guangdong Cocoa Food (40), Jiangsu Lanolin (39) |
| 2018 | Japan (1077), United States (1075), South Korea (899), Malaysia (797), Thailand (682) | Guangdong Vinegar and Vinegar Products (61), Guangdong Vegetable Oil (44), Guangdong Margarine (42), Jiangsu Cocoa Powder (40), Guangdong Knitting Plants (37) |
| 2019 | Japan (1104), United States (1049), South Korea (938), Malaysia (855), Thailand (725) | Guangdong Mineral Water (124), Guangdong Vinegar and Vinegar Products (69), Guangdong Margarine (53), Jiangsu Cocoa Powder (48), Guangdong Vegetable Oil (43) |
| 2020 | Japan (1073), United States (1068), Malaysia (932), South Korea (920), Singapore (721) | Guangdong Vinegar and Vinegar Products (72), Henan Vegetable Wax (55), Guangdong Vegetable Oil (54), Guangdong Margarine (53), Guangdong Cocoa Food (49) |
| 2021 | United States (1111), Japan (1100), Malaysia (1007), South Korea (964), Thailand (713) | Guangdong Vinegar and Vinegar Products (67), Guangdong Margarine (57), Henan Plant Wax (51), Guangdong Plant Oil (49), Guangdong Cocoa Food (44) |
| 2022 | United States (1164), Japan (1131), Malaysia (973), South Korea (969), Vietnam (807) | Guangdong Vinegar and Vinegar Products (75), Guangdong Margarine (55), Guangdong Vegetable Oil (51), Jiangsu Cocoa Powder (48), Henan Plant Wax (43) |

Data source: Calculated by the author based on Chinese Customs data.

export of its agricultural products. From 2015 to 2022, Guangdong's agricultural exports accounted for over 12% of China's total agricultural exports, solidifying its prominence in the sector (Chinese Customs data). The majority of these products are exported to ASEAN countries and those participating in the Belt and Road Initiative.

Other provinces have also demonstrated competitiveness; Shandong Beer, Jiangsu Lanolin, Henan Plant Wax, and Jiangsu Cocoa Powder have gained a footing in the international market. For instance, Shandong Qingdao Beer has prioritized a comprehensive development strategy to improve its domestic and international standing and boost its influence in the international market. This strategy, combined with significant advancements in agricultural information technology, fosters the agricultural economic progress and export growth of Shandong Province [62].

However, the yearly rankings of the top 5 provincial agricultural products in degrees show variability and instability in competitive edge across provinces, except for Guangdong Province. Figs 3 and 4 contrast the export competitiveness networks of the top five products in 2015 and 2022. The red nodes represent provincial agricultural products, and their size is proportional to the degree; the purple nodes represent countries. The edges represent TCA values, and their widths are positively correlated with the TCA values. Notably, only Guangdong Vinegar and Vinegar Products and Guangdong Vegetable Oil have remained constant over the years. Fig 5 depicts the number of agricultural products with competitive advantages that Chinese provinces exported in 2015 and 2022. The figure shows that Shandong, Guangdong, and Zhejiang export the most competitive agricultural products.

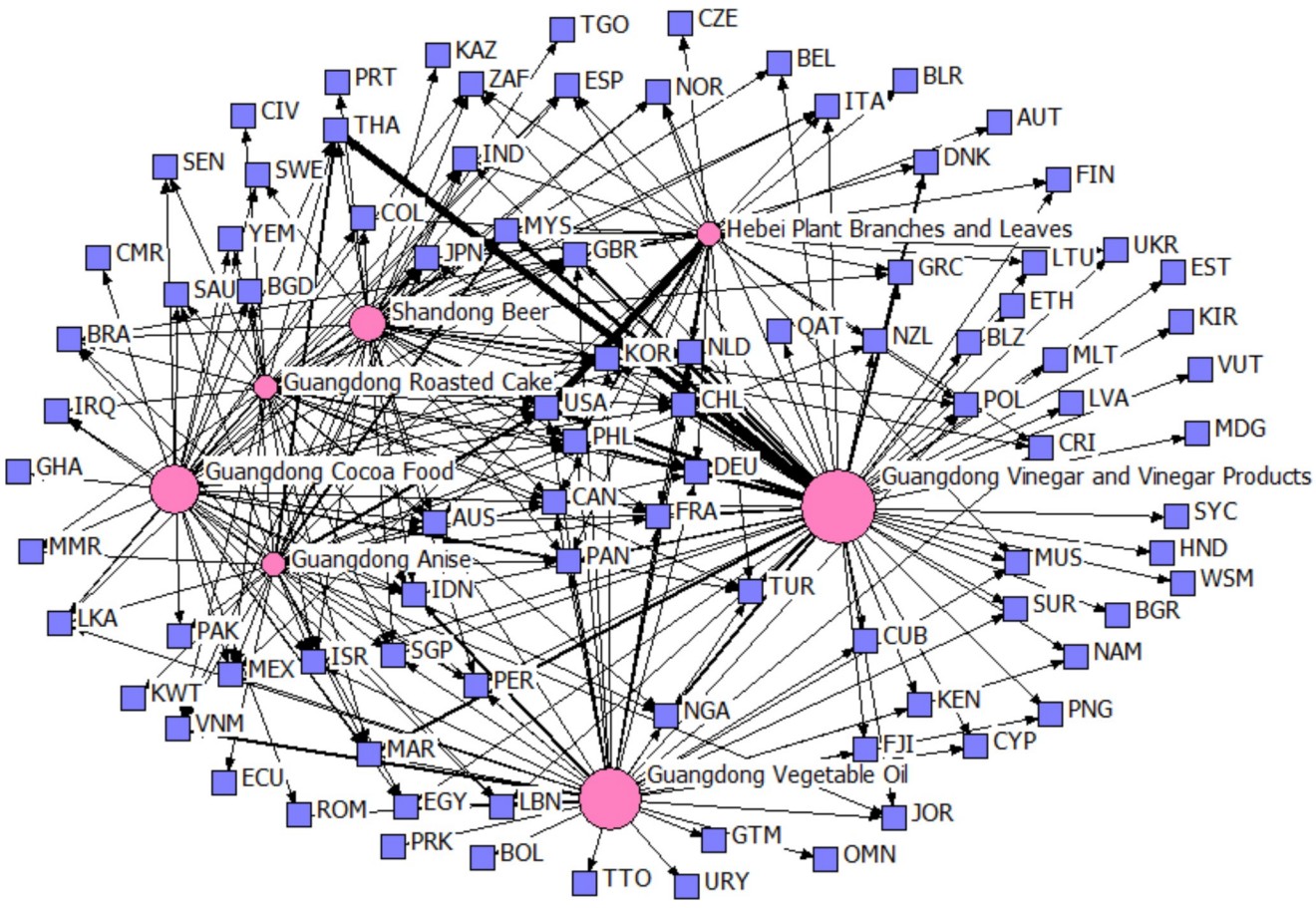

**Fig 3. The export competitiveness network of the top five provincial agricultural products in 2015.** (data source: Chinese Customs data).

## Empirical analysis

### Construction of ERGM and variable explanation

The ERGM is the chosen methodology for examining the factors influencing the establishment of an export competitiveness network of provincial agricultural products in this paper for the following reasons:

1. The ERGM is a crucial empirical regression model utilized in complex network analysis. It helps explain the characteristics of network structure and the factors that influence its formation. The ERGM views the whole network as a combination of local network effects, seeing the observed network as one possible outcome among a set of random networks [63]. Compared to the linear regression methods employed by Long [5] and Bojnec and Ferto [6], which only explore the factors influencing the competitiveness of agricultural exports from an external perspective. The ERGM provides insights into the formation of relationships and the underlying factors that shape them, grounded in the nature and logic of network construction.

2. In contrast to traditional linear regression models, such as those presented by Zhou and Tong [4] and Huyen and Bang [10], ERGM does not exclude the issue of endogeneity. An essential category of explanatory variables in ERGM is network self-organization, which refers to the network structure of the explanatory variables. The underlying rationale is that

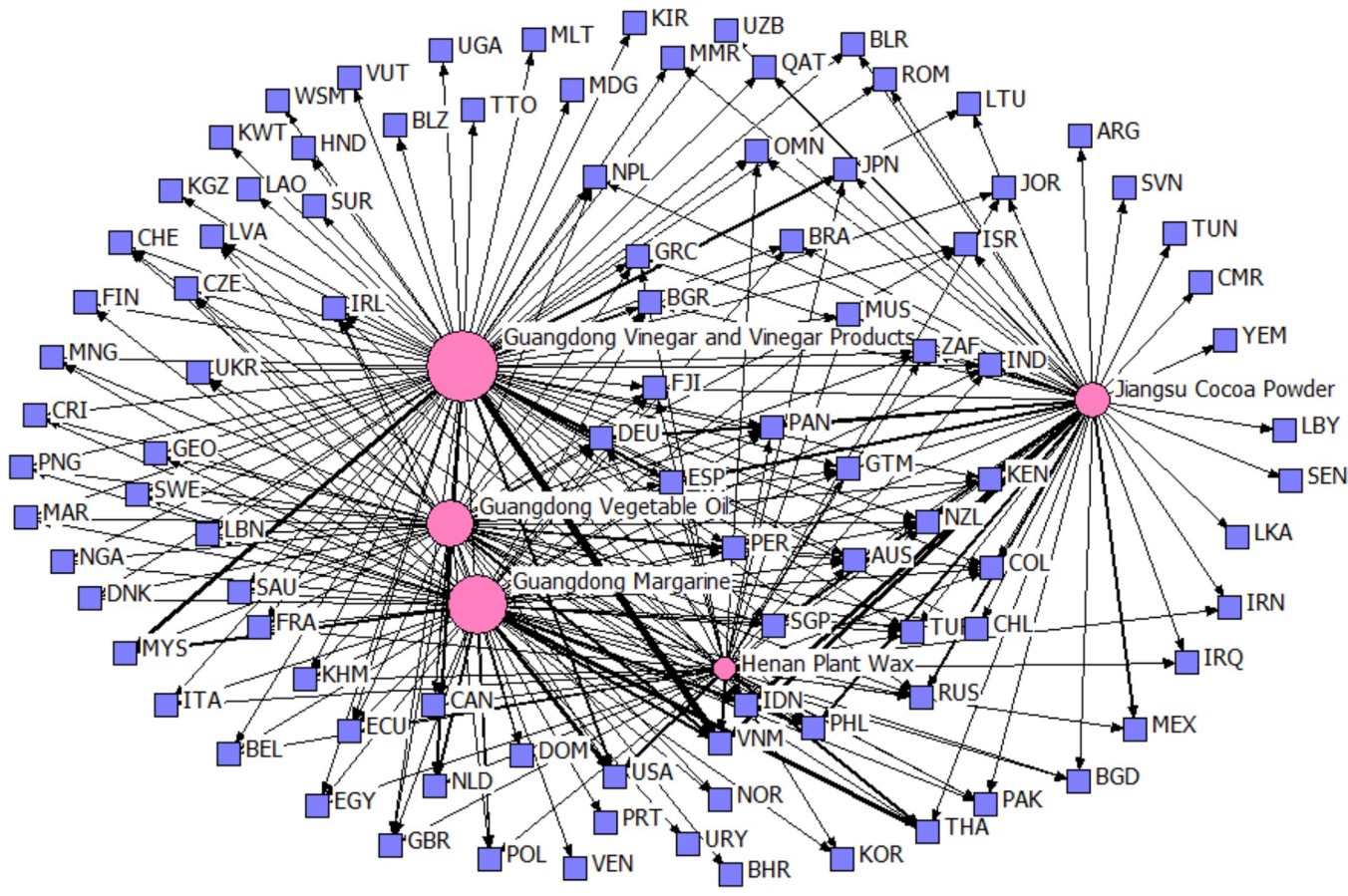

**Fig 4. The export competitiveness network of the top five provincial agricultural products in 2022.** (data source: Chinese Customs data).

network connections are both endogenous and interdependent [22]. Pre-existing network relationships significantly influence the formation of new connections within the network. It is indeed true that established product export relationships will further accumulate experience for the product's overseas expansion and facilitate its export to additional countries. Newly established export relationships are somewhat dependent on existing ones. Thereby, ERGM aims to comprehend the formation of the entire network through its local structure [64].

3. Additionally, the ERGM incorporates three types of explanatory variables: self-organization, node attribute indicators, and external covariate network. Relative to other network models, such as the Quadratic Assignment Procedure (QAP), which focuses on the correlation and regression between matrices [65], this model does not take into account for the effects of node attributes and covariate networks on relationship development. Similarly, the Simulation Investigation for Empirical Network Analysis (SIENA) primarily incorporates explanatory variables related to structural effects and individual actor attribute effects [66]. However, it also fails to consider external covariate networks. Based on the ERGM, we can comprehensively identify critical influences [41] and promote the theoretical hypothesis regarding the formation of export competitiveness networks.

Given its comprehensiveness and practicality, ERGM is widely used as a statistical analysis tool for both one-mode [67,68] and two-mode networks [21,69,70]. Therefore, this paper employs the ERGM.

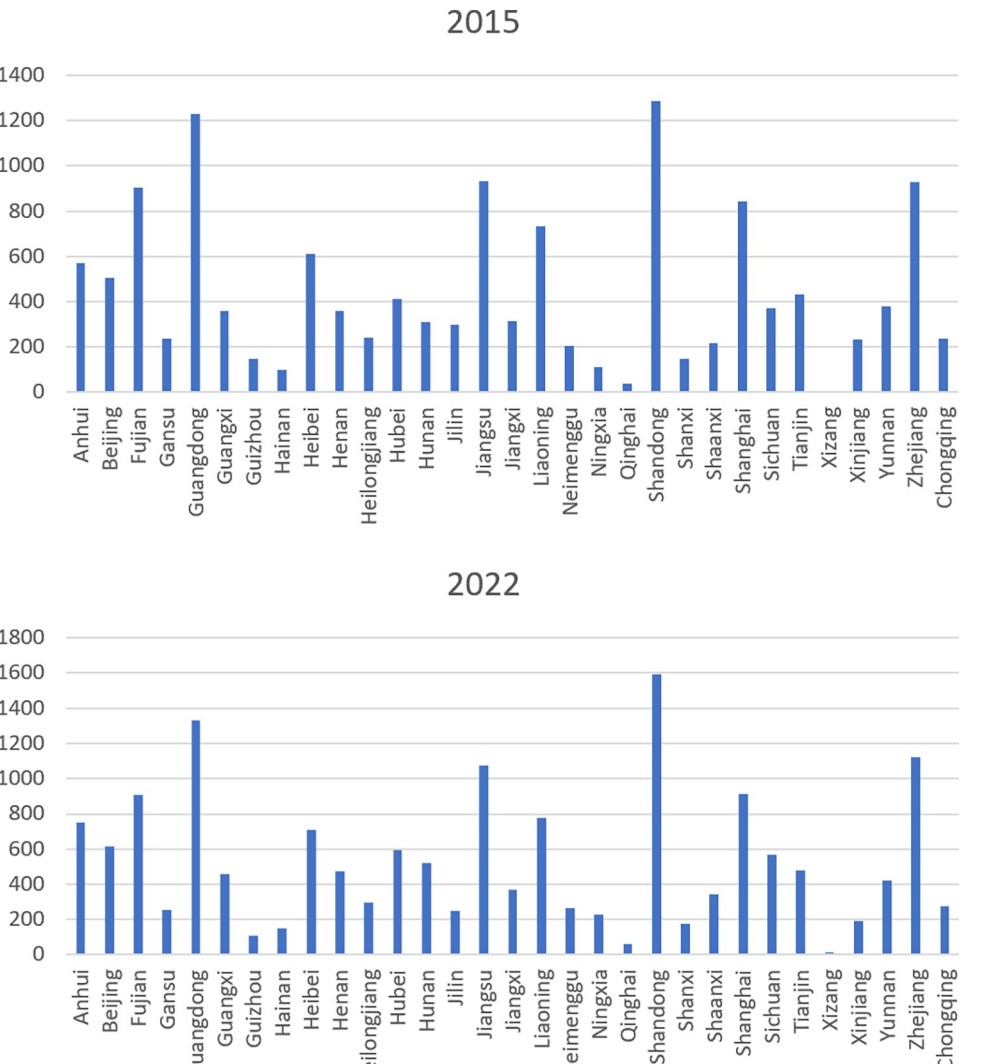

**Fig 5. The number of agricultural products with competitive advantages that Chinese provinces exported in 2015 and 2022.** (data source: Chinese Customs data).

The general form of the ERGM can be expressed as follows:

$$P_{\theta,\gamma}(Y = y|\theta) = \frac{\exp\{\theta^{\mathsf{T}}g(y) + \theta_a^{\mathsf{T}}g(y, X) + \theta_b^{\mathsf{T}}g(y, Z)\}}{\sum_{\eta\in\gamma}\exp\{\theta^{\mathsf{T}}g(\eta)\}}, y \in \gamma \qquad (6)$$

In Eq (6), the dependent variable represents the export competitiveness network of provincial agricultural products. The independent variables include three types of network structure statistics based on theoretical analysis and hypothesis: pure structural effects, node attribute indicators, and external covariate networks.

**Pure structural effects.**   This paper focuses on the pure structural effects, including network edges, 2-star, and 3-star. It aims to investigate the potential for developing new competitive export relationships for provincial agricultural products within the framework of existing competitive export relationships.

**Node attribute indicators.** At the provincial level, this paper accounts for GDP, the total output value of agriculture, forestry, animal husbandry, and fishery (AFAHF), and exports [71]. Each province's GDP and export data for each province are sourced from the China Statistical Yearbook, while the AFAHF data come from the China Rural Statistical Yearbook. Nationally, it is important to account for GDP, population [72], and economic freedom [73] attributes. Each country's GDP and population data are derived from the World Bank's World Development Indicators database, while the Global Heritage Foundation provides data about economic freedom.

**External covariate network.** The international sister-city network (Nfriendcity) plays a significant role. Improved bilateral political relations can enhance the efficiency of agricultural trade and contribute to overall trade effectiveness [74]. Additionally, the model incorporates a physical distance network, Ndist, that measures the distance from provinces to countries, highlighting the substantial effect of spatial distance on trade. Details of the regression variables can be found in Table 3.

This paper employed the stepwise method to perform an analysis aimed at identifying the most suitable regression model using the 2015 data. The star effects of the country were removed as the strong correlation when incorporating the star effects of both node types in the ERGM. As this paper primarily investigates the competitive advantage from the perspective of provincial agricultural products, the model only incorporates the star effect of provincial agricultural products.

**Table 3. Meaning and explanation of ERGM regression variables.**

| Variable | Meaning | Data type | Explanation |
|---|---|---|---|
| **Dependent variable** | | | |
| G | Export competitiveness network | Network data | The export competitiveness network of provincial agricultural product |
| **independent variable** | | | |
| Pure structural effects | | | |
| edges | Edges | Numerical | Intercept |
| 2-star | 2-star | Numerical | Is there a trend toward more competitive export relationships for provincial agricultural produce? |
| 3-star | 3-star | Numerical | |
| Node attribute indicators | | | |
| ProGDP | GDP of provinces | Numerical | Does the ProGDP contribute to the formation of export competitive advantage? |
| Export | Exports of provincial | Numerical | Does Export contribute to the formation of export competitive advantage? |
| AFAHF | The total output value of agriculture, forestry, animal husbandry, and fishery in the province | Numerical | Does AFAHF contribute to the formation of export competitive advantage? |
| CouGDP | GDP of countries | Numerical | Does the CouGDP contribute to the formation of export competitive advantage? |
| MonetaryF | Monetary freedom of countries | Numerical | Does the MonetaryF contribute to the formation of export competitive advantage? |
| TradeF | Trade freedom of countries | Numerical | Does the TradeF contribute to the formation of export competitive advantage? |
| People | People of countries | Numerical | Do people contribute to the formation of a competitive export advantage? |
| External covariate network | | | |
| Nfriendcity | International sister-city network | Network data | Does the Nfriendcity contribute to the formation of export competitive advantage? |
| Ndist | Province-national distance network | Network data | Does the Ndist contribute to the formation of export competitive advantage? |

**Table 4. Gradual regression results of ERGM for the export competitiveness network of Chinese provincial agricultural products (2015).**

| | G | | | | | |
|---|---|---|---|---|---|---|
| | (1) | (2) | (3) | (4) | (5) | (6) |
| edges | -4.287*** (-269.40) | -4.288*** (-281.06) | -4.283*** (-278.16) | -4.383*** (-205.40) | -6.024*** (-73.06) | -6.122*** (-72.45) |
| b1star2 | 0.183*** (59.50) | 0.183*** (62.01) | 0.182*** (61.72) | 0.178*** (57.10) | 0.183*** (61.11) | 0.178*** (58.18) |
| b1star3 | -0.006*** (-25.87) | -0.006*** (-26.51) | -0.006*** (-26.46) | -0.006*** (-25.51) | -0.006*** (-26.17) | -0.006*** (-25.98) |
| Nfriendcity | | 0.245*** (3.30) | 2.065*** (69.98) | 2.024*** (62.04) | 1.988*** (70.56) | 1.934*** (64.65) |
| Ndist | | | -1.150*** (-30.29) | -1.161*** (-28.0) | -1.139*** (-31.24) | -1.134*** (-29.45) |
| ProGDP | | | | -0.012 (-0.67) | | -0.013 (-0.72) |
| AFAHF | | | | 0.047** (2.42) | | 0.046** (2.41) |
| Export | | | | 0.135*** (14.11) | | 0.135*** (14.85) |
| CouGDP | | | | | 0.090*** (5.34) | 0.090*** (5.27) |
| TradeF | | | | | -0.003*** (-4.40) | -0.003*** (-4.31) |
| MonetaryF | | | | | 0.008*** (8.94) | 0.008*** (8.97) |
| People | | | | | 0.344*** (17.09) | 0.344*** (16.88) |
| AIC | 111,990.3 | 111,980.4 | 111,898.0 | 111,593.9 | 110,618.3 | 110,319.5 |
| BIC | 112,022.9 | 112,023.8 | 111,952.3 | 111,680.8 | 110,716.1 | 110,449.9 |

Note

*, **, and *** are significant at 10%, 5%, and 1% significance levels, respectively. The regression coefficients in parentheses correspond to the estimated t-values.

Model (1) solely considers the pure structural effects of provincial agricultural products. The results confirm that the coefficients for the star effects align with the expected outcomes. Models (2) and (3) progressively incorporate the covariate network. Notably, the coefficients for the covariate network correspond with anticipations, and both the AIC and BIC values decrease. Models (4) and (5) gradually incorporate the attributes of provincial agricultural product and country attributes. Upon including these attributes, Model (6) demonstrates the lowest AIC and BIC values. This model's explanatory variables mostly match expectations, hence designating it as the optimal model. The results are shown in Table 4.

## Analysis of regression results

**Basic regression analysis.** Using Model 6 as the baseline model, we next analyze the factors influencing the export competitiveness network of Chinese provincial agricultural products. However, due to a lack of up-to-date attribute data for some provinces and countries in 2022, the regression covers the period from 2015 to 2021. The results can be found in Table 5.

The 2-star with a positive coefficient significantly fosters export competitiveness for agricultural products in the global market, indicating a "rich get richer" effect. In situations where competitive export relationships are already in place, Chinese provincial agricultural products tend to form more. This means that competitive export relationships are continually

**Table 5. Regression results of the ERGM for the export competitiveness network of Chinese provincial agricultural products from 2015 to 2021.**

| | G | | | | | | |
|---|---|---|---|---|---|---|---|
| | **2015** | **2016** | **2017** | **2018** | **2019** | **2020** | **2021** |
| edges | -6.122*** | -4.544*** | -3.520*** | -6.199*** | -3.974*** | -6.085*** | -5.481*** |
| | (-72.45) | (-55.58) | (-49.20) | (-53.83) | (-1,966.86) | (-62.87) | (-70.10) |
| b1star2 | 0.178*** | 0.179*** | 0.174*** | 0.170*** | 0.111*** | 0.149*** | 0.159*** |
| | (58.18) | (59.45) | (60.71) | (61.20) | (190.26) | (60.58) | (64.39) |
| b1star3 | -0.006*** | -0.006*** | -0.006*** | -0.005*** | -0.001*** | -0.004*** | -0.005*** |
| | (-25.98) | (-27.03) | (-27.23) | (-27.17) | (-140.06) | (-26.33) | (-28.14) |
| Nfriendcity | 1.934*** | 1.958*** | 1.882*** | 1.801*** | 1.743*** | 1.740*** | 1.937*** |
| | (64.65) | (71.90) | (69.63) | (71.13) | (264.42) | (71.64) | (79.73) |
| Ndist | -1.134*** | -1.088*** | -1.077*** | -1.000*** | -0.938*** | -0.995*** | -1.131*** |
| | (-29.45) | (-30.25) | (-31.30) | (-30.51) | (-243.09) | (-31.70) | (-37.28) |
| ProGDP | -0.013 | 0.022 | -0.020 | -0.004 | -0.003*** | -0.001 | 0.014 |
| | (-0.72) | (1.33) | (-1.47) | (-0.26) | (-3.28) | (-0.10) | (1.11) |
| AFAHF | 0.046** | 0.052*** | 0.067*** | 0.078*** | 0.041*** | 0.072*** | 0.061*** |
| | (2.41) | (2.95) | (5.03) | (5.69) | (53.32) | (5.66) | (5.17) |
| Export | 0.135*** | 0.123*** | 0.099*** | 0.100*** | 0.046*** | 0.109*** | 0.107*** |
| | (14.85) | (14.46) | (13.26) | (13.78) | (107.63) | (13.80) | (14.70) |
| CouGDP | 0.090*** | -0.009 | 0.220*** | -0.499*** | 0.609*** | -0.188*** | 0.097*** |
| | (5.27) | (-0.50) | (12.73) | (-27.20) | (1,932.15) | (-12.49) | (5.79) |
| TradeF | -0.003*** | 0.030*** | 0.011*** | 0.013*** | 0.017*** | -0.008*** | 0.027*** |
| | (-4.31) | (28.27) | (13.08) | (14.20) | (68.09) | (-12.02) | (27.26) |
| MonetaryF | 0.008*** | -0.029*** | -0.021*** | 0.025*** | -0.034*** | 0.025*** | -0.026*** |
| | (8.97) | (-29.78) | (-31.98) | (21.17) | (-112.79) | (23.97) | (-47.05) |
| People | 0.344*** | 0.003 | -0.245*** | 0.070*** | -0.245*** | 0.338*** | 0.248*** |
| | (16.88) | (0.17) | (-12.33) | (3.44) | (-712.15) | (19.18) | (12.84) |
| AIC | 110,319.5 | 115,334.0 | 122,547.3 | 122,998.9 | 137,338.8 | 128,219.0 | 128,947.5 |
| BIC | 110,449.9 | 115,465.1 | 122,678.7 | 123,130.6 | 137,471.2 | 128,350.3 | 129,079.3 |

Note

*, **, and *** are significant at 10%, 5%, and 1% significance levels, respectively. The regression coefficients in parentheses correspond to the estimated t-values.

expanding. As a result, the 2-star of the export competitiveness network of provincial agricultural products contributes to the formation of network relationships. This paper can, therefore, confirm that the model's results verify H1. Simultaneously, the 3-star coefficient shows a significantly negative value. As trade costs rise with the addition of too many competitive export relationships for agricultural products, provinces are likely to refrain from establishing numerous such relationships. Therefore, a negative 3-star coefficient indicates a "ceiling" on maintained friendships [75].

Social network analysis typically features Markov models with convergence, characterized by positive 2-star coefficients and negative 3-star coefficients [21]. China has emphasized the importance of promoting high-quality agricultural development, expanding favorable agricultural products, and creating new competitiveness for agricultural products in international markets. At the same time, an appropriate number of competitive export relationships is optimal.

AFAHF and Export significantly contribute to establishing the export competitiveness network of provincial agricultural products. The coefficients for AFAHF and Export are positive, and all pass the 5% significance level test. A higher AFAHF suggests a greater likelihood for a province to develop competitive advantages in its agricultural products. Provincial export establishes a channel for distributing agricultural products from each province, facilitating access to international markets. This access has increased the global market share of these

products and fostered competitive advantages. However, the impact of provincial economic development (ProGDP) on establishing competitive advantages in agricultural exports is negligible. Evidence suggests that economically developed provinces may not necessarily exhibit strong competitiveness in agricultural exports.

CouGDP's influence on the competitive advantage of provincial agricultural exports is statistically insignificant. Financially developed countries are likely to receive agricultural exports from various provinces in China. This scenario makes it difficult for provincial agricultural products to form a competitive edge in financially developed export markets. Furthermore, the coefficient for trade freedom (TradeF) is significantly positive, indicating that higher trade freedom equals lower tariffs and tariff barriers. This reduction in trade costs enhances the accessibility of Chinese agricultural products in the host market. Consequently, the importance of promoting high-level agricultural liberalization to ensure seamless exports from China has risen. Therefore, these findings offer partial support to H2.

The positive coefficient for Nfriendcity signifies the beneficial role of international sister-city networks in establishing export competitiveness for Chinese provincial agricultural products. These networks act as a valuable resource of social capital, providing natural advantages in enhancing bilateral trade and driving economic development and partnerships [76]. Specifically, by lowering trade costs, sharing market information and consumer preferences, and conducting agricultural cooperation projects, the sister-city networks ultimately help build competitive advantages for agricultural exports. Consequently, playing the role of a sister-city platform has become a crucial strategy for many provinces and cities to promote their agricultural products. This finding supports H3.

Conversely, the coefficient for the distance network (Ndist) shows a steadily negative trend. Rising trade costs associated with distance can hinder the formation of competitive export relationships for agricultural products across China's provinces. The result confirms H4.

**Robustness analysis.** This paper employs one-period lagged data and replacement regression methods for robustness testing. Regression analyses of the export competitiveness network of Chinese provincial agricultural products from 2016 to 2022 are conducted based on a lagged-by-one-period external covariate network and node attribute indicators. The corresponding regression results are presented in Table 6.

The results demonstrate that the coefficients for pure structural effects and the lagged-by-one-period external covariate network closely align with the benchmark regression results, passing the 1% significance test. Besides, the coefficients for Export also match the benchmark regression results. Hence, the regression utilizing the lagged-by-one-period external covariate network provides robust outcomes.

The fixed effects model was employed for robustness testing. The results demonstrate that the coefficients of key variables-namely, international sister-city (lnfriendcity), bilateral distance (lndist), and provincial export (lnExport)-are consistent with the benchmark regression. Thus, the regression results remain robust even after altering the regression method.

**Heterogeneity analysis.** The influencing factors and effects may vary significantly among provinces due to differences in geographic location, resource endowment, economic development, and other factors. This paper divides the 31 provinces into three regions-eastern, central, and western-based on their province location for the heterogeneity analysis.

The formation of competitive advantages for provincial agricultural exports is uniformly influenced by pure structural effects and external covariate networks. In the eastern and central provinces, where the sister-city relationship plays a limited role in boosting the competitiveness of agricultural exports, the coefficients are generally lower than those of the western provinces. These findings emphasize the importance of considering the province's economic

**Table 6. Robustness test.**

| | G | | | | lnG | |
|---|---|---|---|---|---|---|
| | 2016 | 2018 | 2020 | 2022 | plm | |
| edges | -4.648*** | -5.153*** | -6.403*** | -4.459*** | lnfriendcity | 0.006*** |
| | (-57.38) | (-50.52) | (-66.70) | (-56.44) | | (4.51) |
| b1star2 | 0.179*** | 0.170*** | 0.149*** | 0.154*** | lndist | -0.651*** |
| | (59.67) | (56.67) | (74.50) | (77.00) | | (-20.42) |
| b1star3 | -0.006*** | -0.005*** | -0.004*** | -0.004*** | lnProGDP | -0.030 |
| | (-25.82) | (-24.95) | (-27.68) | (-25.19) | | (-0.34) |
| Nfriendcity | 1.871*** | 1.750*** | 1.765*** | 1.892*** | lnAFAHF | 0.001 |
| | (8.54) | (9.56) | (10.09) | (10.87) | | (0.02) |
| Ndist | -0.999*** | -0.930*** | -1.007*** | -1.042*** | lnExport | 0.084*** |
| | (-6.17) | (-6.74) | (-7.40) | (-7.78) | | (2.69) |
| ProGDP | 0.010 | -0.071 | -0.003 | 0.061 | lnCougdp | -0.007 |
| | (0.14) | (-1.08) | (-0.04) | (0.97) | | (-0.21) |
| AFAHF | 0.053 | 0.106*** | 0.084** | 0.037 | lnTradeF | -0.004 |
| | (1.47) | (3.31) | (2.63) | (1.42) | | (-0.20) |
| Export | 0.121*** | 0.124*** | 0.107*** | 0.083*** | lnMonetaryF | 0.003 |
| | (5.26) | (5.39) | (4.12) | (3.46) | | (0.18) |
| CouGDP | -0.058*** | -0.467*** | -0.208*** | 0.098*** | lnPeople | 0.834* |
| | (-3.41) | (-25.94) | (-13.87) | (6.53) | | (1.78) |
| TradeF | 0.031*** | 0.022*** | -0.008*** | 0.002** | Constant | 18.687 |
| | (29.45) | (21.73) | (-12.38) | (2.85) | | (1.20) |
| MonetaryF | -0.029*** | 0.004*** | 0.029*** | 0.007*** | FE | Yes |
| | (-29.06) | (3.97) | (28.32) | (8.42) | Observations | 107,846 |
| People | 0.054** | 0.004 | 0.358*** | -0.293*** | R2 | 0.084 |
| | (2.70) | (0.20) | (19.89) | (-17.24) | Adjusted R2 | 0.082 |
| AIC | 115328.207 | 123294.792 | 128068.574 | 139770.993 | Residual S. E. | 2.280 (df = 107643) |
| BIC | 115459.298 | 123426.433 | 128199.926 | 139903.760 | F Statistic | 48.778***(df = 202; 107643) |

Note

*, **, and *** are significant at 10%, 5%, and 1% significance levels, respectively. The regression coefficients in parentheses correspond to the estimated t-values.

development and openness when exploring the factors that influence the competitiveness of agricultural exports.

The impact of node attributes on forming competitive advantages in agricultural product exports across different regions reveals the heterogeneity. The CouGDP positively influences the competitiveness of agricultural product exports in the eastern provinces but has a negative effect in the central and western provinces. Agricultural products from the eastern provinces benefit from lower transportation costs, making them more competitive in developed markets. In contrast, the central and western provinces tend to export agricultural products to neighboring countries due to transportation cost constraints.

Trade freedom negatively impacts the competitive advantage of eastern provinces in agricultural product exports while benefiting western provinces. Despite fostering product exports, trade freedom hinders the establishment of competitive relationships in agricultural products due to an over-supply in high-freedom trade markets. Since the western provinces must consider transportation costs, they prefer to export their competitive agricultural products to countries with open trade policies. Finally, a country's population does not significantly influence the development of competitive advantage in agricultural product exports in eastern

**Table 7. Regression results considering the division of eastern, central, and western provinces.**

| | G | | | | | |
|---|---|---|---|---|---|---|
| | Eastern provinces | | Central provinces | | Western provinces | |
| | **2015** | **2021** | **2015** | **2021** | **2015** | **2021** |
| edges | -4.954*** | -3.275*** | -6.751*** | -7.131*** | -6.217*** | -4.981*** |
| | (-55.95) | (-40.54) | (-27.90) | (-32.39) | (-32.39) | (-30.62) |
| b1star2 | 0.166*** | 0.145*** | 0.187*** | 0.153*** | 0.204*** | 0.194*** |
| | (43.50) | (46.78) | (18.34) | (23.83) | (22.98) | (26.74) |
| b1star3 | -0.005*** | -0.004*** | -0.007*** | -0.004*** | -0.008*** | -0.007*** |
| | (-19.53) | (-21.07) | (-6.84) | (-9.44) | (-8.91) | (-10.95) |
| Nfriendcity | 1.407*** | 1.544*** | 1.176** | 1.080*** | 1.737*** | 1.834*** |
| | (48.65) | (64.70) | (2.45) | (2.71) | (4.54) | (6.40) |
| Ndist | -0.711*** | -0.810*** | -0.498* | -0.437* | -0.742*** | -0.757*** |
| | (-16.70) | (-24.43) | (-1.78) | (-1.89) | (-2.95) | (-3.91) |
| AFAHF | 0.015 | 0.015 | 0.808*** | -1.498*** | 0.430*** | 0.071 |
| | (0.16) | (0.19) | (11.54) | (-17.47) | (2.76) | (0.67) |
| ProGDP | 0.056 | 0.071** | -0.906*** | 2.540*** | -0.375** | 0.046 |
| | (1.27) | (2.29) | (-14.50) | (35.73) | (-2.01) | (0.24) |
| Export | 0.102*** | 0.099*** | 0.118 | 0.394** | 0.062 | -0.032 |
| | (3.52) | (3.21) | (0.95) | (2.21) | (0.83) | (-0.55) |
| CouGDP | 0.317*** | 0.128*** | -0.723*** | -0.504*** | -0.345*** | -0.192*** |
| | (15.22) | (6.97) | (-16.84) | (-15.11) | (-8.78) | (-5.44) |
| TradeF | -0.004*** | -0.006*** | -0.003** | 0.001 | 0.008*** | 0.023*** |
| | (-4.80) | (-7.52) | (-2.05) | (0.59) | (4.24) | (10.89) |
| MonetaryF | -0.008*** | -0.007*** | 0.023*** | 0.019*** | 0.0004 | -0.022*** |
| | (-8.20) | (-8.40) | (8.61) | (8.31) | (0.19) | ((-18.02) |
| People | 0.201*** | -0.154*** | 0.927*** | 0.640*** | 0.661*** | 0.326*** |
| | ((8.06) | ((-7.15) | ((18.47) | ((16.36) | ((13.91) | ((8.13) |
| AIC | 67,941.7 | 81,103.5 | 18,480.0 | 23,026.5 | 22,979.3 | 27,468.6 |
| BIC | 68,064.3 | 81,227.8 | 18,591.2 | 23,138.7 | 23,094.2 | 27,584.6 |

Note

*, **, and *** are significant at 10%, 5%, and 1% significance levels, respectively. The regression coefficients in parentheses correspond to the estimated t-values.

provinces. However, it has a noticeable impact in central and western provinces. The results presented in Table 7.

This paper further classifies countries as those along the Belt and Road, compared to those not along the Belt and Road, conducting an in-depth heterogeneous analysis (see Table 8).

The international sister-city network shows a significant positive influence in both categories. However, the coefficients for the international sister-city network are lower for countries along the Belt and Road. Friendly relations between Chinese provinces and countries along the Belt and Road are not entirely mirrored in the agricultural export competitiveness. Ultimately, the establishment of competitive agricultural export relationships by Chinese provinces along the Belt and Road is also shaped by contractual relationships, which diminish the influence of sister-city relations. Thus, there is a need for further examination into the role of international sister-city relationships in fostering the competitive advantages of provincial agricultural exports to countries along the Belt and Road. Underpinned by a combination of national and local political relations, it is easier for Chinese provinces to establish competitively advantageous agricultural export relationships.

Only provincial exports significantly influence the competitive advantage of agricultural exports to two categories of countries. In contrast, the impacts of other variables, such as provincial and country GDP and trade freedoms, vary and lack clear patterns.

**Table 8. Regression results considering the division of countries along the Belt and Road and non-Belt and Road.**

| | G | | | | | |
| --- | --- | --- | --- | --- | --- | --- |
| | Countries along the Belt and Road | | | Countries along the non-Belt and Road | | |
| | 2015 | 2018 | 2021 | 2015 | 2018 | 2021 |
| edges | -6.154*** | 1.219*** | -8.040*** | -6.876*** | -4.952*** | -7.543*** |
| | (-50.75) | (7.33) | (-39.63) | (-65.36) | (-44.05) | (-76.93) |
| b1star2 | 0.403*** | 0.346*** | 0.279*** | 0.249*** | 0.261*** | 0.301*** |
| | (28.22) | (34.01) | (35.12) | (42.74) | (42.77) | (48.36) |
| b1star3 | -0.035*** | -0.025*** | -0.015*** | -0.012*** | -0.014*** | -0.015*** |
| | (-12.02) | (-14.23) | (-13.63) | (-18.06) | (-18.71) | (-20.33) |
| Nfriendcity | 1.114*** | 1.103*** | 1.238*** | 2.772*** | 3.134*** | 2.321*** |
| | (3.27) | (4.78) | (4.86) | (55.22) | (77.01) | (54.12) |
| Ndist | -0.536*** | -0.461*** | -0.636*** | -1.949*** | -1.998*** | -1.779*** |
| | (-2.76) | (-3.04) | (-3.66) | (-35.28) | (-44.63) | (-38.10) |
| ProGDP | -0.075 | -0.082 | 0.117 | 0.212** | -0.154 | 0.045 |
| | (-0.44) | (-0.59) | (0.90) | (2.05) | (-1.51) | (0.47) |
| AFAHF | 0.286*** | 0.287*** | 0.157*** | -0.088* | 0.122*** | 0.029 |
| | (3.07) | (4.39) | (2.890) | (-1.80) | (2.61) | (0.75) |
| Export | 0.272*** | 0.192*** | 0.134*** | 0.122*** | 0.203*** | 0.147*** |
| | (4.96) | (3.94) | (2.69) | (3.60) | (5.64) | (4.06) |
| CouGDP | 0.588*** | 0.449*** | 0.189*** | -0.235*** | -0.617*** | 2.045*** |
| | (13.30) | (12.95) | (6.57) | (-11.16) | (-29.06) | (77.07) |
| TradeF | 0.037*** | -0.00004 | 0.009*** | 0.017*** | 0.031*** | -0.027*** |
| | (19.45) | (-0.04) | (7.32) | (11.70) | (22.70) | (-27.92) |
| MonetaryF | -0.047*** | -0.058*** | 0.019*** | -0.015*** | -0.013*** | -0.005*** |
| | (-25.60) | (-36.60) | (9.11) | (-11.08) | (-14.44) | (-5.20) |
| People | 0.159*** | -0.814*** | 0.365*** | 0.967*** | 0.285*** | -0.518*** |
| | (4.36) | (-22.18) | (11.81) | (33.46) | (11.79) | (-16.47) |
| AIC | 39,772.7 | 49,484.0 | 54,231.9 | 67,873.8 | 73,678.1 | 61,577.8 |
| BIC | 39,891.0 | 49,603.8 | 54,351.8 | 67,998.8 | 73,804.1 | 61,704.1 |

Note

*, **, and *** are significant at 10%, 5%, and 1% significance levels, respectively. The regression coefficients in parentheses correspond to the estimated t-values.

# Research conclusions and policy implications

## Conclusions

As China increasingly engages with the global community, cultivating partnerships between Chinese and international agricultural sectors enables countries to utilize their comparative advantages. This strategy paves the way for innovative, sustainable, and high-quality development of global agriculture. This study constructs a competitiveness network of agricultural product exports in Chinese provinces, utilizing export data from 2015 to 2022. The goal is to evaluate the competitive performance of provincial agricultural product exports in the international market and to explore the factors influencing this competitiveness network.

Our analysis leads to several conclusions. Firstly, the competitive edge of agricultural exports from various Chinese provinces needs enhancement. Secondly, aside from Guangdong Province, the stability of competitively advantaged agricultural exports is uncertain in other provinces. Thirdly, the international sister-city network aids the development of export competitiveness networks for agricultural products from Chinese provinces, while geographical distance hinders the formation of competitive export trade relationships. Lastly, the creation of this export competitiveness network is also influenced by provincial and national characteristics, suggesting a degree of heterogeneity.

## Policy implications

Based on the research findings, this paper argues that the government should prioritize the establishment of a high-quality development strategy for agricultural products in the provinces of China, focusing on four key aspects:

(1) Enhancing and developing agricultural products with traditional advantages in Chinese provinces while strengthening collaboration with research institutes and relevant enterprises. Specifically, provincial governments can improve agricultural products through standardized, large-scale, and modernized production models, thereby further strengthening the advantages of traditional agricultural products. On this basis, they can create distinctive and competitive agricultural products in each province based on their agricultural diversity. For example, Shandong Province can emphasize its strengths in exporting vegetables and fruits, Zhejiang Province can strengthen the international competitiveness of its tea industry, and Yunnan Province can focus on building its strengths in flower exports, among other initiatives. Specifically, the government can increase funding and concentrate on fostering scientific and technological innovation within agribusinesses. It is essential to establish a high-quality team of experts in the field of agriculture, consisting of faculty and students from universities, as well as agricultural scientists. Furthermore, emphasis should be placed on fostering collaboration among local governments, agricultural universities, and agriculture-related institutions and enterprises.

(2) Provincial governments should actively create agricultural brands and form a competitive group of geographical indications for agricultural products to stabilize the superior agricultural products.

On the one hand, provincial governments should provide policy support and guidance to integrate modern agricultural industrial parks and advantageous specialty industrial clusters, thereby creating agricultural product brands with strong competitive advantages. It is worthwhile to learn from the experiences of the wine industry cluster in California. California's wine sector has achieved high intensification, characterized by mechanized grape harvesting, a well-developed integrated transportation system, and a global distribution network. As a result, operational efficiency and brand competitiveness have significantly improved. On the other hand, local governments should leverage major agribusinesses as the foundation for agricultural industrialization to achieve economies of scale.

(3) Provincial governments must make the most of diplomatic relations at the local level to build a platform for agricultural cooperation. The approach taken by Guizhou Province in fostering friendly relations and signing an agreement for international agricultural cooperation with Luang Namtha Province in Laos is commendable. It is essential to leverage sister-city relationships to create new opportunities for bilateral economic and trade cooperation and to enhance the competitiveness of provincial agricultural exports. International agricultural cooperation agreements between provinces and global partners should be encouraged to increase awareness of provincial agricultural products through bilateral cultural and economic exchanges. At the same time, special attention must be paid to exploring complementary spaces for agricultural products between Chinese provinces and countries along the Belt and Road.

(4) Different regions should carry out reasonable regional planning for agricultural products to avoid competition of homogenized agricultural products in the same market. The eastern region should fully utilize its geographic advantages by targeting the United States, the European Union, Japan, South Korea, and other countries as the main export destinations for its advantageous agricultural products. Meanwhile, the central and western regions should actively promote the diversification of agricultural export markets based on the China Railway

Express and the new western land-sea corridor. By differentiating the layout of agricultural products in different markets, it is easier to form a competitive advantage of agricultural products.

## Limitations

However, several deficiencies still exist in this paper and need further research.

First, ERGM is primarily used to empirically analyze the influence of relationship establishment. However, future research should focus on empirically analyzing the impact of relationship strength on the export competitiveness network of Chinese provincial agricultural products.

Furthermore, the data in the China Customs database has been available since 2015, making the current timeframe under study relatively short. A longer data span can enhance statistical credibility and improve the generalizability of the findings. Therefore, it is essential to consider examining longer time series data in future studies.

Moreover, while this paper incorporates three types of variables in the empirical regression analysis, some of these variables are not statistically significant. Future research will explore additional variables at the levels of Chinese provinces and countries to enhance its validity.

## Supporting information

**S1 File. Provincial agricultural exports.** Procedural command file.
(R)

**S2 File. Data file.**
(RAR)

## Acknowledgments

We would like to thank the editor and the anonymous reviewers for their helpful suggestions and comments.

## Author Contributions

**Conceptualization:** Ye Chen, Zhuoxi Liu, Zhongming Yin.

**Data curation:** Ye Chen.

**Formal analysis:** Ye Chen, Zhongming Yin.

**Funding acquisition:** Ye Chen.

**Methodology:** Ye Chen.

**Writing – original draft:** Ye Chen, Zhuoxi Liu.

**Writing – review & editing:** Ye Chen.

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
