## [Decision Letter · Decision Letter 0]

1 Apr 2024

PONE-D-24-04543Research on the evolution of the structure and influencing factors of the export competitiveness network of Chinese provincial agricultural products

——Based on a two-mode network perspectivePLOS ONE

Dear Dr. Chen,

Thank you for submitting your manuscript to PLOS ONE. After careful consideration, we feel that it has merit but does not fully meet PLOS ONE’s publication criteria as it currently stands. Therefore, we invite you to submit a revised version of the manuscript that addresses the points raised during the review process.

We look forward to receiving your revised manuscript.

Kind regards,

Tinggui Chen

Academic Editor

PLOS ONE

Journal Requirements:

"Social Science Planning Project of Sichuan Province: Research on the Comparative Advantage Structure Evolution and Development Strategies of Sichuan Province's Agricultural Products Export under the Belt and Road Initiative (SC22C012)."

**Additional Editor Comments:**

I have completed my evaluation of your manuscript. The reviewers recommend reconsideration of your manuscript following major revision. I invite you to resubmit your manuscript after addressing the comments below.

Reviewers' comments:

Reviewer's Responses to Questions

**Comments to the Author**

1. Is the manuscript technically sound, and do the data support the conclusions?

Reviewer #1: Yes

Reviewer #2: Yes

Reviewer #3: Yes

Reviewer #4: Yes

2. Has the statistical analysis been performed appropriately and rigorously? 

Reviewer #1: Yes

Reviewer #2: Yes

Reviewer #3: Yes

Reviewer #4: Yes

3. Have the authors made all data underlying the findings in their manuscript fully available?

Reviewer #1: Yes

Reviewer #2: No

Reviewer #3: No

Reviewer #4: No

4. Is the manuscript presented in an intelligible fashion and written in standard English?

Reviewer #1: Yes

Reviewer #2: Yes

Reviewer #3: No

Reviewer #4: No

5. Review Comments to the Author

Reviewer #1: Greetings! This manuscript reveals the foreign trade network of agricultural products at the provincial level, which is an important guide for sustainable regional agricultural development. However, this article needs more in-depth analysis. I suggest the following three aspects for improvement:

(1)Some of the content of the article lacks support from the literature, especially in the introduction. In the introduction, the authors only cite a large number of policy documents to highlight the practical significance of this study, but the scientific significance is largely not mentioned.

(2)The authors should discuss the key findings obtained from the study.

(3)Authors should enhance the condensation of the conclusion and abstract sections rather than simply repeating important findings.

Reviewer #2: The manuscript, titled "Research on the evolution of the structure and influencing factors of the export competitiveness network of Chinese provincial agricultural products – Based on a two-mode network perspective," has examined the changes in agricultural exports and the factors that impact their export competitiveness.

This is an excellent paper, and it deals with an important and timely issue using quantitative analytical tools. They have rightly used network analysis. However, the authors must address the following issues to enhance the paper's quality, readability, and communication.

•Title of the paper: The title of the manuscript is very long. The title of an article is very important to enhance its searchability, reach the audience, engage readers, influence their thoughts, and increase the probability of citations. The title is expected to be understandable to people, even those who are not experts in the field. Therefore, to reflect the content and make it easily understandable, the title may be revised as "Export competitiveness network of Chinese provincial agricultural products: Evolution, performance, and influencing factors."

•Strengthen the discussion section: The authors cited other studies, mainly in the introduction and methodology section, to highlight the importance of examining an issue. They have not compared the results of the present study with other studies dealing with similar topics, such as the export competitiveness of agricultural products (for example, Long, 2020) and the influence of Belt and Road on competitive advantage (for example, Feng et al., 2020). They can also cite the results of studies on trade competitiveness and international sister cities. For example, the manuscript has cited the study by Ramasamy and Cremer (1998), who analyzed the economic role of international sister-city relationships between New Zealand and Asia. Readers and policymakers will appreciate it if the authors cite the results of such studies. Thoughtful comparison always enhances the quality and impact of our research.

References

Feng, L., Xu, H., Wu, G., Zhao, Y., & Xu, J. (2020). Exploring the structure and influence factors of trade competitive advantage network along the Belt and Road. Physica A: Statistical Mechanics and Its Applications, 559, 125057.

Long, Y. (2021). Export competitiveness of agricultural products and agricultural sustainability in China. Regional Sustainability, 2(3), 203-210.

Ramasamy, B., & Cremer, R. D. (1998). Cities, commerce and culture: The economic role of international sister‐city relationships between New Zealand and Asia. Journal of the Asia Pacific Economy, 3(3), 446-461.

Reviewer #3: The paper has assessed the competitive advantage position of provincial agricultural product exports in the international market and investigated the influencing factors of the competitiveness network of provincial agricultural product exports.

1. Does the citation format of the literature meet the requirements of the journal? The author needs to confirm it again.

2. In introduction, the main research problems and research contributions of this study should be clearly defined.

3. It is suggest the authors to add a discussion on the application research of the dual mode network.

4. The figures from this manuscript are not clear. Please ask the authors to resolve this issue.

5. Does Guangxi produce coconut? Please ask the authors to confirm it.

6. The models required for this study should be placed in one section

7. The robust analysis should be added.

8. More robust analysis should be provided.

9. Finally, this manuscript needs careful editing by someone with expertise in technical English editing.

Reviewer #4: Dear Authors,

Your research on the evolution of the export competitiveness network of Chinese provincial agricultural products and its influencing factors offers valuable insights into a critical area of agricultural economics and international trade. After a thorough review, I would like to provide some constructive feedback to enhance the clarity, depth, and impact of your work.

1. Originality and Significance:

Your study makes a significant contribution by using export customs data from 2015 to 2022 to analyze the competitive network of agricultural exports across various Chinese provinces. The focus on the dynamic evolution of this network and the systematic analysis of influencing factors is commendable. However, it would be beneficial to more explicitly state how this work builds upon and diverges from existing literature in the field. Highlighting the novel aspects of your research will underscore its contribution to academic and practical knowledge.

2. Theoretical Framework:

The paper could benefit from a clearer articulation of the theoretical framework guiding the analysis. While the empirical approach is robust, integrating a theoretical perspective on networks in international trade or theories of competitive advantage could provide a stronger foundation for interpreting the results. This addition would also help in situating your work within broader academic discussions.

3. Methodology:

The methodology section is well-developed, leveraging export customs data to construct a network analysis of provincial agricultural product competitiveness. However, details on the specific network analysis techniques and statistical methods used would enhance the reader's understanding and the reproducibility of the study. Additionally, discussing any limitations of your methodological approach would lend credibility to your findings.

4. Results and Discussion:

Your findings regarding the increasing competitiveness of provincial agricultural exports, the need for enhanced competitive advantages, the role of international sister city relationships, and the influence of agricultural output, exports, and trade freedom indexes are intriguing. It would be valuable to deepen the discussion on how these factors interplay and their implications for policy and practice. Exploring the heterogeneity in the effects of node attributes across different provincial regions and export destinations in more detail could provide nuanced insights into strategic planning for export competitiveness.

5. Policy Implications and Recommendations:

The policy recommendations provided are a strong point of the paper, particularly the emphasis on cultivating distinctive and globally competitive agricultural products and leveraging the Belt and Road initiative. Expanding on how these strategies can be implemented at both the provincial and national levels, considering the varied economic and agricultural contexts across China, would make the recommendations more actionable.

6. Writing Quality and Organization:

Overall, the manuscript is well-organized and clearly written. Ensuring consistency in terminology and further proofreading to catch any grammatical or typographical errors would polish the final manuscript. It may also be helpful to include more visual aids, such as charts or graphs, to illustrate key findings and trends.

In conclusion, your manuscript provides a valuable contribution to the field of agricultural economics and international trade. With the suggested revisions, I believe the paper will offer stronger insights and clearer guidance for researchers, policymakers, and practitioners interested in enhancing the export competitiveness of Chinese provincial agricultural products. I look forward to seeing the revised manuscript and believe it has the potential to make a significant impact.

6. PLOS authors have the option to publish the peer review history of their article (what does this mean?). If published, this will include your full peer review and any attached files.

Reviewer #1: No

Reviewer #2: **Yes: **Uttam Deb

Reviewer #3: No

Reviewer #4: No

---

## [Author Response · Author response to Decision Letter 0]

12 Jun 2024

We wish to thank the reviewer 1 for her/his detailed comments and suggestions that helped to tremendously improve the manuscript. 

Reviewer #1: Greetings! This manuscript reveals the foreign trade network of agricultural products at the provincial level, which is an important guide for sustainable regional agricultural development. However, this article needs more in-depth analysis. I suggest the following three aspects for improvement:

(1) Some of the content of the article lacks support from the literature, especially in the introduction. In the introduction, the authors only cite a large number of policy documents to highlight the practical significance of this study, but the scientific significance is largely not mentioned.

We noticed that there is a lengthy discussion about policy documents. Thus, we made deletions and added explanations of scientific significance.

In the Introduction section, the policy documents paragraph was rephrased as followed:

In 2022, China’s Ministry of Agriculture and Rural Affairs rolled out the “14th Five Year Plan for International Cooperation in Agriculture and Rural Affairs”, emphasising the enhancement of international collaboration in agriculture and the development of competitive advantages in agricultural trade. This focus on competitive advantage has now become a significant influencing factor in China’s agricultural cooperation endeavours [1].

The explanation of scientific significance was added in the Introduction:

Despite this progress, Chinese agriculture contends with challenges such as limited resources, land fragmentation, and high production costs [2]. These issues have impacted the competitiveness of China’s agricultural exports, which have revealed a comparative disadvantage [3,4], hindering the export growth and sustainable development of the sector [5]. In light of these circumstances, it is imperative to assess the current situation, identify the reasons, and understand the factors influencing the competitiveness of China’s agricultural product exports. This will aid in shaping a high-quality development model for China’s agricultural trade within the existing development framework.

(2) The authors should discuss the key findings obtained from the study.

We thank the reviewer for raising this important point. We have made some additions and deletions to the discussion on the research conclusions.

1)In the 5.Network structure characteristics and evolution analysis section, the following phrase was added:

It is clear that provinces selectively opt for specific export markets for their agricultural products to cultivate a competitive advantage. However, this advantage in competitiveness falls under 4% across the 174 countries (regions) analysed in this paper. Thus, the competitiveness of Chinese provincial agricultural exports is essentially minimal.

The following phrase was deleted:

The process of choosing export markets for agricultural products can create a degree of exclusivity, establishing a competitive advantage and potentially reducing or avoiding competition among different agricultural products in the same international market. As a result, it will facilitate the establishment of a competitive edge in global markets and serve as a demonstration base for agricultural exports from the province.

So, the first conclusion is:

Firstly, there are significant opportunities to improve the competitive edge of agricultural product exports across various Chinese provinces.

2) In the 5.Network structure characteristics and evolution analysis section, the following phrase was added:

However, the yearly rankings of the top 5 agricultural products show variability and instability across provinces, with only Guangdong Province consistently showing a competitive edge.

So, the second conclusion is:

Secondly, with the exception of Guangdong Province, the types of competitively robust agricultural products exported by each province are inconsistent. 

3)Based on the research findings of the paper, the third conclusion is:

Thirdly, international sister-city relationships can foster the development of an export competitiveness network of Chinese provincial agricultural products, whereas geographic distance may hinder this network.

4) the fourth conclusion is:

Lastly, the development of this competition-focused export network is also influenced by province-level and national (or regional) characteristics, and there are unique differences within these influences.

(3) Authors should enhance the condensation of the conclusion and abstract sections rather than simply repeating important findings.

We apologize for the redundancy concerning the statements in the conclusion and abstract sections of the paper. We have sorted out the article's conclusions and extracted key conclusions.

In the conclusion and abstract section, the corresponding paragraph was rephrased as followed:

Our analysis leads to several conclusions. Firstly, the competitive edge of agricultural exports from various Chinese provinces needs bolstering. Secondly, aside from Guangdong Province, the stability of competitively advantaged agricultural exports is uncertain in other provinces. Thirdly, the international sister-city network aids the development of Chinese provincial agricultural product export competitiveness, whereas distance hinders the formation of competitive export trade relationships. Lastly, the creation of this export competitiveness network is also influenced by provincial and national (regional) characteristics, suggesting a degree of heterogeneity.

We are grateful for the constructive comments of reviewer 2 that helped to strengthen the manuscript.

Reviewer #2: The manuscript, titled "Research on the evolution of the structure and influencing factors of the export competitiveness network of Chinese provincial agricultural products – Based on a two-mode network perspective," has examined the changes in agricultural exports and the factors that impact their export competitiveness.

This is an excellent paper, and it deals with an important and timely issue using quantitative analytical tools. They have rightly used network analysis. However, the authors must address the following issues to enhance the paper's quality, readability, and communication.

•Title of the paper: The title of the manuscript is very long. The title of an article is very important to enhance its searchability, reach the audience, engage readers, influence their thoughts, and increase the probability of citations. The title is expected to be understandable to people, even those who are not experts in the field. Therefore, to reflect the content and make it easily understandable, the title may be revised as "Export competitiveness network of Chinese provincial agricultural products: Evolution, performance, and influencing factors."

Research on the evolution of the structure and influencing factors of the export competitiveness network of Chinese provincial agricultural products – Based on a two-mode network perspective

As proposed by the reviewer, we adopted revised title as the title of our paper:

Export competitiveness network of Chinese provincial agricultural products: Evolution, performance, and influencing factors.

•Strengthen the discussion section: The authors cited other studies, mainly in the introduction and methodology section, to highlight the importance of examining an issue. They have not compared the results of the present study with other studies dealing with similar topics, such as the export competitiveness of agricultural products (for example, Long, 2020) and the influence of Belt and Road on competitive advantage (for example, Feng et al., 2020). They can also cite the results of studies on trade competitiveness and international sister cities. For example, the manuscript has cited the study by Ramasamy and Cremer (1998), who analyzed the economic role of international sister-city relationships between New Zealand and Asia. Readers and policymakers will appreciate it if the authors cite the results of such studies. Thoughtful comparison always enhances the quality and impact of our research.

We apologize for not having compared the results of the present study with other studies dealing with similar topics. As suggested, we added literature comparison and rephrased the respective sentences in the corresponding section.

In the 2. Literature review section, the paragraph was rephrased as followed:

In summary, while current literature provides extensive theoretical and empirical research on the competitive advantage of agricultural trade and trade networks, it lacks a focused, detailed look at the export competitiveness of agricultural products from individual Chinese provinces. Some studies have examined the export competitiveness of Chinese agricultural products [5], but none have assessed the influencing factors and evolutionary patterns of such competitiveness via a network perspective. This gap leaves us without a comprehensive understanding of provincial agricultural product export networks.

In the 4.1 Network construction section, the second paragraph was rephrased as followed: 

Feng et al. [52] primarily researched RCA networks along the Belt and Road without categorising the products. These fall under one-mode networks. We examine the export competitiveness network of different agricultural products from a more granular, provincial viewpoint, which is classified as a two-mode network. This micro two-mode network is particularly suitable for analysing the export competitiveness of Chinese provinces’ agricultural products.

We are thankful to reviewer 3 for his valuable comments that helped to profoundly improve the manuscript.

Reviewer #3: The paper has assessed the competitive advantage position of provincial agricultural product exports in the international market and investigated the influencing factors of the competitiveness network of provincial agricultural product exports.

1. Does the citation format of the literature meet the requirements of the journal? The author needs to confirm it again.

We thank the reviewer for raising this important point. Based on the requirements and specifications of the journals regarding formatting, we used endnote software to fully adjust the formatting of references.

2. In introduction, the main research problems and research contributions of this study should be clearly defined.

As suggested, we rephrased the research problems and research contributions of this study in the Introduction section. 

The innovations in this study are primarily seen in several aspects. First, using provincial agricultural product export data, we categorise agricultural product types based on the 4-digit Harmonized System (HS). This allows for a detailed exploration of the current status and issues concerning the global reach of Chinese provincial agricultural products. Second, applying a two-mode network perspective and an enhanced revealed comparative advantage index, we innovatively construct an export competitiveness network of provincial agricultural products. This provides insight into the network’s evolution, growth, and the current state of provincial export competitiveness for agricultural products in China. Third, by using the Exponential Random Graph Model (ERGM), we investigate the factors influencing the development of the export competitiveness network of provincial agricultural product and offer sound policy suggestions for nurturing competitive advantages in provincial agricultural products.

3. It is suggest the authors to add a discussion on the application research of the dual mode network.

We thank the reviewer for raising this important point. As suggested, we added a discussion on the application research of the two-mode network.

Research on the two-mode network

Initial research into two-mode networks emphasised clustering, density, and centrality metrics [23]. Since then, indicators of structural features and empirical econometric models have been developed [24,25]. By adopting dynamic thinking, researchers have evolved the study of two-mode networks from a static to a dynamic perspective [26]. Various practical applications of two-mode networks such as firm-city networks [27,28], debt-bonding networks between non-financial and financial listed companies [29], networks mapping Southern women’s activity participation [30], stakeholder and cost management networks [8], and fund-stock networks [31], have been thoroughly explored. Such studies typically encompass the conversion of a two-mode network into a one-mode network, the structural assessment of a two-mode network, and the analysis of factors influencing a two-mode network.

4. The figures from this manuscript are not clear. Please ask the authors to resolve this issue.

We apologize that the graphs are blurry in the paper. The figures in the paper are as follows:

Fig. 1. A schematic diagram of the construction of a two-mode network for provincial agricultural exports.

(a)

(b)

Fig 2. The export competitiveness networks of the top five provincial agricultural products in 2015 (a) and 2022 (b). (data source: Chinese customs data). Note: The red nodes represent provincial agricultural products, and their size is proportional to the degree; the purple nodes represent countries or regions. The edges represent TCA values, and their widths are positively correlated with the TCA values.

(a)

(b)

Fig 3. The number of agricultural products with competitive advantages that Chinese provinces exported in 2015 (a) and 2022 (b). (data source: Chinese customs data).

5. Does Guangxi produce coconut? Please ask the authors to confirm it.

As suggested, we went further and checked the data. In a report on the development of the coconut industry, it is noted that coconuts are mainly distributed in Hainan, Yunnan, Guangdong, Guangxi, Fujian, Taiwan and other provinces and regions of China. Among them, Hainan is the main production area. We looked further into Chinese customs data. In 2022, a total of 11 provinces in China exported coconuts, and only Guangxi province exported coconuts to the United States. Therefore, Guangxi produces coconut.

6. The models required for this study should be placed in one section

We thank the reviewers for this suggestion. Two parts of this paper include formulas. One, in Chapter 4, analyses the network by constructing the network and the network metrics. This part of the formulas aims to perform a network structure analysis to describe the development process of constructing the network. Secondly, the formulas are involved in the model construction part of Chapter Five. This part is based on Chapter 4. They are differentiated. Generally speaking, the research paradigm using network analysis methodology includes network structure analysis and empirical regression analysis, with different research contents and perspectives. After much deliberation, placing the two parts of the formula in the same section is inappropriate.

7. The robust analysis should be added.

We thank the reviewer for pointing to this important methodological aspect of our paper. We have now added robust analysis in the Analysis of regression results section.

Robustness analysis

This paper employs lagged data and replacement regression methods for robustness testing. The corresponding regression results can be found in Table 6.

Table 6. Robustness test.

 G lnG

 2016 2018 2020 2022 plm

edges -4.648*** -5.153*** -6.403*** -4.459*** lnfriendcity 0.006***

 (-57.38) (-50.52) (-66.70) (-56.44) (4.51)

b1star2 0.179*** 0.170*** 0.149*** 0.154*** lndist -0.651***

 (59.67) (56.67) (74.50) (77.00) (-20.42)

b1star3 -0.006*** -0.005*** -0.004*** -0.004*** lnProGDP -0.030

 (-25.82) (-24.95) (-27.68) (-25.19) (-0.34)

Nfriendcity 1.871*** 1.750*** 1.765*** 1.892*** lnAFAHF 0.001

 (8.54) (9.56) (10.09) (10.87) (0.02)

Ndist -0.999*** -0.930*** -1.007*** -1.042*** lnExport 0.084***

 (-6.17) (-6.74) (-7.40) (-7.78) (2.69)

ProGDP 0.010 -0.071 -0.003 0.061 lnCougdp -0.007

 (0.14) (-1.08) (-0.04) (0.97) (-0.21)

AFAHF 0.053 0.106*** 0.084** 0.037 lnTradeF -0.004

 (1.47) (3.31) (2.63) (1.42) (-0.20)

Export 0.121*** 0.124*** 0.107*** 0.083*

---

## [Decision Letter · Decision Letter 1]

1 Jul 2024

PONE-D-24-04543R1Export competitiveness network of Chinese provincial agricultural products: Evolution, performance, and influencing factorsPLOS ONE

Dear Dr. Chen,

Thank you for submitting your manuscript to PLOS ONE. After careful consideration, we feel that it has merit but does not fully meet PLOS ONE’s publication criteria as it currently stands. Therefore, we invite you to submit a revised version of the manuscript that addresses the points raised during the review process.

We look forward to receiving your revised manuscript.

Kind regards,

Tinggui Chen

Academic Editor

PLOS ONE

Journal Requirements:

Reviewers' comments:

Reviewer's Responses to Questions

**Comments to the Author**

1. If the authors have adequately addressed your comments raised in a previous round of review and you feel that this manuscript is now acceptable for publication, you may indicate that here to bypass the “Comments to the Author” section, enter your conflict of interest statement in the “Confidential to Editor” section, and submit your "Accept" recommendation.

Reviewer #2: All comments have been addressed

Reviewer #3: All comments have been addressed

Reviewer #4: All comments have been addressed

2. Is the manuscript technically sound, and do the data support the conclusions?

Reviewer #2: Yes

Reviewer #3: Yes

Reviewer #4: Yes

3. Has the statistical analysis been performed appropriately and rigorously? 

Reviewer #2: Yes

Reviewer #3: Yes

Reviewer #4: Yes

4. Have the authors made all data underlying the findings in their manuscript fully available?

Reviewer #2: Yes

Reviewer #3: Yes

Reviewer #4: Yes

5. Is the manuscript presented in an intelligible fashion and written in standard English?

Reviewer #2: Yes

Reviewer #3: Yes

Reviewer #4: No

6. Review Comments to the Author

Reviewer #2: Thank you for addressing the comments and suggestions in the revised version of the manuscript. Now, it is an excellent paper with potential impacts for enhancing export competitiveness.

Reviewer #3: The revision has been well improved based on the comments of the reviewers. But, the manuscript still needs some minor improvements based on following comments.

1. The understanding of the research innovations from the authors is biased. The three research innovations mentioned in this paper seem to be the main focus of this study.

2. The current literature lacks a focused, detailed look at the export competitiveness of agricultural products from individual Chinese provinces. But, the authors did not mention the necessity of this research in the introduction,

3. In Empirical analysis section, there are still relatively little comparative analysis with other methods and literature.

4. The main conclusions and policy implications do not seem to correspond one-to-one. Meantime, the policy implications should have specific implementation entities.

Reviewer #4: The authors have largely addressed the suggestions made by the reviewers, but there are still the following details that need attention:

1. Pay attention to formatting, such as aligning the ends of the text and keeping the charts and graphs on the same page.

2. Simplify the content, such as the findings need to be further refined and summarised to make them more logical and organised.

3. describe in detail the construction process of the agricultural export competitiveness network, including the definition of network nodes and edges, data processing methods and other content.

4. carefully proofread and revise the language of the paper to ensure smooth and accurate language expression and avoid grammatical errors and spelling mistakes.

7. PLOS authors have the option to publish the peer review history of their article (what does this mean?). If published, this will include your full peer review and any attached files.

Reviewer #2: **Yes: **Uttam Deb

Reviewer #3: No

Reviewer #4: No

---

## [Author Response · Author response to Decision Letter 1]

30 Jul 2024

We are thankful to reviewer 3 and reviewer 4 for their valuable comments that helped to improve the manuscript once again.

Reviewer #3: The revision has been well improved based on the comments of the reviewers. But, the manuscript still needs some minor improvements based on following comments.

1. The understanding of the research innovations from the authors is biased. The three research innovations mentioned in this paper seem to be the main focus of this study.

We thank the reviewer for raising this important point. We rephrased the research innovations in the Introduction section.

(1) Most existing literature studies the export competitiveness of Chinese agricultural products from a national perspective. This paper, however, connects Chinese provinces with exported agricultural products and examines the international competitiveness of Chinese provincial agricultural products from a more micro perspective.(2) By applying a two-mode network perspective and an enhanced revealed comparative advantage index, this paper innovatively constructs an export competitiveness network of provincial agricultural products, which further enriches the research paradigm and methodology on the trade relationship. (3) Using the Exponential Random Graph Model (ERGM), this paper investigates the factors influencing the development of the export competitiveness network of provincial agricultural products from a comprehensive perspective. Compared to the traditional linear regression model, ERGM can better track the endogenous dependencies in forming the export competitiveness network, a task the conventional regression model cannot accomplish.

2. The current literature lacks a focused, detailed look at the export competitiveness of agricultural products from individual Chinese provinces. But, the authors did not mention the necessity of this research in the introduction.

We apologize for not mentioning the necessity of the export competitiveness of agricultural products from individual Chinese provinces. As suggested, we added a paragraph in the Introduction section.

 The main objective of this study is to investigate the status and influence of Chinese provincial export competitiveness for agricultural products from the perspective of complex networks. The study aims to explore the evolution of the export competitiveness network of Chinese provincial agricultural products and to identify the factors affecting the competitive advantages of agricultural exports at the provincial level. The current literature lacks a focused and detailed exploration of the export competitiveness of agricultural products from Chinese provinces. This study fills this gap by conducting a comprehensive analysis from micro and macro perspectives.

3. In Empirical analysis section, there are still relatively little comparative analysis with other methods and literature.

We thank the reviewer for raising this important point. We have made some additions to the comparative analysis with other methods and literature.

In contrast to traditional linear regression models, such as those in Zhou and Tong [4] and Huyen and Bang [10], ERGM does not exclude the endogeneity problem. Additionally, the model incorporates three different levels of explanatory variables and is capable of identifying key influences as comprehensively as possible [39]. Therefore, ERGM is widely used as a statistical analysis tool for both one-mode [58,59] and two-mode networks [25,60,61].

4. The main conclusions and policy implications do not seem to correspond one-to-one. Meantime, the policy implications should have specific implementation entities.

As proposed by the reviewer, we adjusted the policy recommendations based on our conclusions one by one and indicated the implementing entity for each policy.

The first policy implication corresponds to the first conclusion.

(1) Enhancing and developing agricultural products with traditional advantages in China's provinces, strengthening cooperation with research institutes and relevant enterprises, and cultivating distinctive and high-quality varieties will help increase their international competitiveness and influence. Specifically, the provincial government can improve the quality of agricultural products through standardized, large-scale, and modernized production models and create distinctive and competitive agricultural products in each province based on economic and agricultural diversity. 

The second policy implication corresponds to the second conclusion.

(2) Provincial governments should actively create agricultural brands and form a group of competitive geographical indications for agricultural products to stabilize the superior agricultural products in each province. On the one hand, provincial governments should provide policy support and guidance to integrate modern agricultural industrial parks and advantageous specialty industrial clusters and create agricultural product brands with strong competitive advantages in the provinces. On the other hand, provincial governments can facilitate better exports of advantageous agricultural products by building stable and mature export routes.

The third policy implication corresponds to the third conclusion.

(3) Provincial governments must make the most of diplomatic relations at the local level to build a platform for agricultural cooperation. Leverage international sister-city relationships to create new possibilities for bilateral economic and trade cooperation and to enhance the competitive advantage of provincial agricultural exports in friendly countries. Encourage interactions and collaborations between provinces and global partners to increase awareness and recognition of provincial agricultural products through bilateral cultural and economic exchanges. At the same time, special attention should be paid to exploring complementary spaces for agricultural products between Chinese provinces and countries along the Belt and Road.

The fourth policy implication corresponds to the fourth conclusion.

(4) Different regions should carry out reasonable regional layouts and planning for agricultural products to avoid homogenization and the situation of many but not strong. The eastern region should fully utilize its geographic location to consolidate the dominant position of agricultural products in markets such as the United States, the European Union, Japan, and South Korea, while the central and western regions should actively promote the diversification of agricultural export markets based on the CHINA RAILWAY Express and the new western land-sea corridor.

Reviewer #4: The authors have largely addressed the suggestions made by the reviewers, but there are still the following details that need attention.

1. Pay attention to formatting, such as aligning the ends of the text and keeping the charts and graphs on the same page.

We thank the reviewers for this suggestion. The latest manuscript has been set up to align the ends of the text and keep the charts and graphs on the same page.

2. Simplify the content, such as the findings need to be further refined and summarised to make them more logical and organised.

We thank the reviewer for pointing to this important issue. We rephrased the sentences in the analysis of regression results.

The content of paragraph ' In terms of pure structural effects ' and ' The 2-star with a positive coefficient significantly fosters ' has been simplified based on internal logic. 

The 2-star with a positive coefficient significantly fosters export competitiveness for agricultural products in the global market, indicating a “rich get richer” effect. This means that competitive export relationships are continually expanding. In situations where such relationships are already in place, Chinese provincial agricultural products tend to form more. This paper can, therefore, confirm that the model’s results verify H1. Simultaneously, the 3-star coefficient shows a significantly negative value. As trade costs rise with the addition of too many competitive export relationships for agricultural products, provinces will avoid forming numerous such relationships. 

Social network analysis typically features Markov models with convergence, characterized by positive 2-star coefficients and negative 3-star coefficients [25]. A positive 2-star coefficient suggests a propensity for forming new friendships, while a negative 3-star coefficient implies potential drawbacks from establishing and sustaining a large number of friendships, indicating a “ceiling” on maintained friendships [66].China has emphasized the importance of promoting high-quality agricultural development, paying more attention to expanding favorable agricultural products in international markets, and creating new competitive advantages in exporting specialty agricultural products.

The paragraph of ' Secondly, concerning node attribute variables ' was rephased as followed:

AFAHF and export significantly contribute to establishing the provincial agricultural products’ export competitiveness network. A higher provincial AFAHF suggests a propensity for a province to develop competitive advantages in its agricultural products. Provincial export establishes a channel for agricultural products in each province.

The paragraph of ' The effect of national (or regional) attributes on the competitive advantage ' was rephased as followed:

CouGDP's influence on the competitive advantage of provincial agricultural exports is statistically insignificant. Financially developed countries or regions are more probable recipients of agricultural exports from China's various provinces. This scenario makes it difficult for each province's agricultural exports to form a competitive edge in the financially developed export markets.

The part ' China’s varying economic development and openness across its eastern, central, and western regions can lead to different factors influencing the competitive advantages of agricultural exports in each province. ' was deleted.

The paragraph of ' The regression results in Table 7 indicate that ' was rephased as followed:

The regression results in Table 7 indicate that the formation of competitive advantage for provincial agricultural product exports is influenced uniformly by structural effects and covariate networks. Due to the advanced economy in the eastern and central provinces, where the sister-city relationship has a limited role in boosting their agricultural exports’ competitiveness, the eastern and central provinces generally have lower coefficients than the western provinces. These findings emphasize the importance of considering the provincial’s economic development and openness when exploring the actors influencing the competitive advantages of agricultural exports.

The paragraph ' The coefficient for the international sister-city network ' and ' Ultimately, the establishment of competitive agricultural export relationships ' has been merged as one paragraph based on the content and internal logic. 

The international sister-city network shows a significant positive relationship within both categories. However, this relationship is less pronounced for countries along the Belt and Road. This result reveals that the friendly relations between Chinese provinces and countries (regions) along the Belt and Road are not entirely mirrored in the provinces’ agricultural export competitiveness. Ultimately, the establishment of competitive agricultural export relationships by Chinese provinces along the Belt and Road route is also shaped by contractual relationships, reducing the influence of sister-city relations. Most countries along the Belt and Road possess natural geographical advantages, substantial economic development benefits, and a strong reliance on foreign trade. Thus, the establishment of sister-city relationships holds the potential to enhance cooperation and development further.

3. describe in detail the construction process of the agricultural export competitiveness network, including the definition of network nodes and edges, data processing methods and other content.

We apologize for not describing in detail the construction process of the agricultural export competitiveness network in the paper. As suggested, we have made some additions.

The part ' The network comprises two types of nodes ' was changed as followed: 

The network comprises two types of nodes: one is a composite node encompassing a province and a specific type of agricultural product, meaning each node corresponds to a province and the agricultural product it exports. For example, if Guangdong Province exports Vegetable Oil, then Guangdong-Vegetable Oil constitutes a node. This paper identified a total of 189 agricultural product export categories classified at the HS 4-digit level. The theoretical maximum number of nodes in this type is 189 x 31, combining 31 provinces. The other type of node represents the export destination, signifying a country or region. The edges of the two-mode network of provincial agricultural product exports are the trade relationships established between the province's agricultural products and countries.

The paragraph ' Feng et al. [52] primarily researched RCA networks ' was added a sentence: 

Based on the trade volume of various agricultural products exported from Chinese provinces to different countries, the TCA is constructed as follows:

4. carefully proofread and revise the language of the paper to ensure smooth and accurate language expression and avoid grammatical errors and spelling mistakes.

As suggested, we checked the whole manuscript once again and corrected the misrepresentations.

---

## [Decision Letter · Decision Letter 2]

26 Aug 2024

PONE-D-24-04543R2Export competitiveness network of Chinese provincial agricultural products: Evolution, performance, and influencing factorsPLOS ONE

Dear Dr. Chen,

Thank you for submitting your manuscript to PLOS ONE. After careful consideration, we feel that it has merit but does not fully meet PLOS ONE’s publication criteria as it currently stands. Therefore, we invite you to submit a revised version of the manuscript that addresses the points raised during the review process.

We look forward to receiving your revised manuscript.

Kind regards,

Tinggui Chen

Academic Editor

PLOS ONE

Additional Editor Comments :

I have completed my evaluation of your manuscript. The reviewers recommend reconsideration of your manuscript following major revision. I invite you to resubmit your manuscript after addressing the comments below.

Reviewers' comments:

Reviewer's Responses to Questions

**Comments to the Author**

1. If the authors have adequately addressed your comments raised in a previous round of review and you feel that this manuscript is now acceptable for publication, you may indicate that here to bypass the “Comments to the Author” section, enter your conflict of interest statement in the “Confidential to Editor” section, and submit your "Accept" recommendation.

Reviewer #4: (No Response)

Reviewer #5: All comments have been addressed

2. Is the manuscript technically sound, and do the data support the conclusions?

Reviewer #4: Partly

Reviewer #5: Yes

3. Has the statistical analysis been performed appropriately and rigorously? 

Reviewer #4: Yes

Reviewer #5: Yes

4. Have the authors made all data underlying the findings in their manuscript fully available?

Reviewer #4: No

Reviewer #5: Yes

5. Is the manuscript presented in an intelligible fashion and written in standard English?

Reviewer #4: No

Reviewer #5: Yes

6. Review Comments to the Author

Reviewer #4: The manuscript presents an interesting analysis of the export competitiveness network of Chinese provincial agricultural products from 2015 to 2022. The use of a two-mode network perspective and the Exponential Random Graph Model (ERGM) adds novelty to the research. However, there are a few points that need clarification and improvements to strengthen the manuscript.

1. Introduction Section:

The introduction could benefit from a more concise summary of the research gap, particularly international outlook, and the significance of studying export competitiveness at the provincial level. Clearly stating why this study is needed and how it differs from existing literature will strengthen the motivation. The authors mention three research innovations, but these could be framed more clearly in the context of the overall study goals and objectives.

2. Methodology: Provide more details on the construction of the export competitiveness network. Clarify the definition of network nodes and edges, including the data sources and processing methods. This will help readers replicate the analysis. Explain the choice of ERGM over other statistical models, highlighting its advantages in capturing endogenous dependencies.

3. Empirical Analysis: Strengthen the comparative analysis with other methods and literature. Discuss how ERGM compares to traditional linear regression models in terms of explaining the development of the export competitiveness network. Provide more interpretation of the regression results, particularly the significance of various network effects and covariates. Link the findings back to the research questions and hypotheses.

4. Results and Discussion: The findings could be organized more logically and concisely. Avoid repetition and streamline the explanations to make them more accessible to readers. Discuss potential limitations of the study and suggestions for future research. For example, consider examining longer time series data or including additional covariates.

5. Policy Implications: The policy implications are well-aligned with the conclusions, but could be framed more concretely with specific actions provinces could take. Consider including real-world examples or case studies to illustrate the implementation of these policies.

6. Formatting and Language: Ensure the text is consistently formatted, with aligned ends and charts/graphs kept on the same page. Carefully proofread the manuscript to correct any grammatical errors, spelling mistakes, or unclear phrasing.

Reviewer #5: The manuscript can be considered for publication after the following questions and recommendations:

1-The researchers did not indicate what is new and distinctive in the abstract of this manuscript (compared to other previous studies).

2-The authors did not specify the method used to determine the sample size.

3-The idea of the manuscript is based on the exported products to determine which products have the most significant impact on the global market. Therefore, the Pareto chart will help draw a clearer picture for the reader and summarize the manuscript in one chart. The Pareto chart aims to highlight the most important products among a (usually large) group of products.

4-The researchers did not mention why this regression method was used to analyze the results.

7. PLOS authors have the option to publish the peer review history of their article (what does this mean?). If published, this will include your full peer review and any attached files.

Reviewer #4: No

Reviewer #5: No

---

## [Author Response · Author response to Decision Letter 2]

20 Sep 2024

We are thankful to reviewer 4 and reviewer 5 for their valuable comments that helped to improve the manuscript.

Reviewer #4: The manuscript presents an interesting analysis of the export competitiveness network of Chinese provincial agricultural products from 2015 to 2022. The use of a two-mode network perspective and the Exponential Random Graph Model (ERGM) adds novelty to the research. However, there are a few points that need clarification and improvements to strengthen the manuscript.

1. Introduction Section:

The introduction could benefit from a more concise summary of the research gap, particularly international outlook, and the significance of studying export competitiveness at the provincial level. Clearly stating why this study is needed and how it differs from existing literature will strengthen the motivation. The authors mention three research innovations, but these could be framed more clearly in the context of the overall study goals and objectives.

We thank the reviewer for raising this important point. As suggested, we have made some additions and rephrased portions of the Introduction section.

The third paragraph in the Introduction section has been supplemented with additional sentences: 

Notably, the primary competitive agricultural products in the global market include wheat from Russia, soybeans from the United States, and rice from Thailand. As a major agricultural nation, China is relatively deficient in globally competitive agricultural products.

The first sentence in the fourth paragraph was rephrased as follows:

The competitive advantage of exported products is crucial for the high-quality economic development of a country or region.

We have added a sentence in the fourth paragraph:

Due to variations in geographic location and resource endowment, China's exported agricultural products exhibit distinct geographical characteristics and imbalances.

A sentence in the fourth paragraph was rephrased as follows:

Therefore, it is essential to provide complete insights into the agricultural export advantages of each province [8] and to identify the factors that propel the effective international expansion of these advantageous agricultural products [9,10].

A sentence in the fifth paragraph was rephrased as follows:

Exploring the evolution of the export competitiveness network of Chinese provincial agricultural products and identifying the influencing factors will help promote the high-quality development of agricultural trade in China's provinces.

A sentence in the sixth paragraph was rephrased as follows:

This paper, however, connects Chinese provinces with their exported agricultural products and examines the international competitiveness of Chinese provincial agricultural products from a more micro perspective to further explain the regionalization and differentiation characteristics of competitive agricultural products across Chinese provinces.

A sentence has been added to the sixth paragraph:

This approach more effectively identifies agricultural products with competitive advantages in each province and further enriches the research paradigm and methodology regarding the trade relationship.

2. Methodology: Provide more details on the construction of the export competitiveness network. Clarify the definition of network nodes and edges, including the data sources and processing methods. This will help readers replicate the analysis. Explain the choice of ERGM over other statistical models, highlighting its advantages in capturing endogenous dependencies.

We apologize for not describing in detail the construction process of the agricultural export competitiveness network in the paper. 

This paper develops the export competitiveness network of provincial agricultural products based on the two-mode network for provincial agricultural exports. These two types of networks have the same nodes but different edges. As suggested, we enhanced the description of the nodes and edges within the export competitiveness network of Chinese provincial agricultural products in the part of Network construction.

Therefore, the nodes of the export competitiveness network of Chinese provincial agricultural products include two categories: the composite nodes of agricultural products exported from Chinese provinces and the destination countries. The edges of the export competitiveness network are the TCA index with a value greater than 1.5.

As suggested, we have made some additions about data sources and processing methods in Network construction.

We added a sentence in the first paragraph of the Network construction:

The data was sourced from China Customs data, which has been available since 2015. By 2023, obtaining complete data for the year ending in 2022 may be possible. Therefore, the research sample in this paper is based on data from 2015 to 2022 to ensure consistency and coherence.

We added a sentence in the second paragraph of the Network construction:

We excluded the sample of countries with missing data on economic freedom for more than two consecutive years, resulting in 186 countries. From this group, we further excluded countries that had missing GDP data for more than two consecutive years. Ultimately, the final sample comprised 174 countries.

A sentence about why we determine the sample size based on HS 4 to classify provincial agricultural products has been added to the second paragraph of the Network construction:

The advantage of classifying agricultural products at the HS 4-digit level is that it effectively distinguishes between various agricultural products without leading to an excessively granular and diverse range. In this paper, we have excluded the Hong Kong, Macao, and Taiwan regions of China.

We summarize the reasons for selecting the ERGM based on three key points, the second of which illustrates the advantages in capturing endogenous dependencies.

(1) The ERGM is a crucial empirical regression model utilized in complex network analysis. It helps explain the characteristics of network structure and the factors that influence its formation. The ERGM views the whole network structure as a combination of local network effects, seeing the observed network as one possible outcome among a set of random networks [56]. Compared to the linear regression methods employed in Long [5] and Bojnec and Ferto [6], which only explore the factors influencing the competitiveness of agricultural exports from an external perspective, the ERGM provides insights into the formation of relationships and the underlying factors that shape them, grounded in the nature and logic of network construction.

(2) In contrast to traditional linear regression models, such as those presented by Zhou and Tong [4] and Huyen and Bang [10], ERGM does not exclude the issue of endogeneity. An essential category of explanatory variables in ERGM is network self-organization, which refers to the network structure of the explanatory variables. The underlying rationale is that network connections are both endogenous and interdependent [35]. Pre-existing network relationships significantly influence the formation of new connections within the network. Thereby, ERGM aims to comprehend the formation of the entire network through its local structure [57]. 

(3) Additionally, the model incorporates three different types of explanatory variables: self-organization, node attribute indicators, and external covariate network. Based on this, we can comprehensively identify critical influences [39] and promote the theoretical hypothesis regarding the formation of export competitiveness networks.

3. Empirical Analysis: Strengthen the comparative analysis with other methods and literature. Discuss how ERGM compares to traditional linear regression models in terms of explaining the development of the export competitiveness network. Provide more interpretation of the regression results, particularly the significance of various network effects and covariates. Link the findings back to the research questions and hypotheses.

In explaining the choice of ERGM over other statistical models, we have made some additions to compare ERGM with traditional linear regression models in the Construction of ERGM and variable explanation:

Compared to the linear regression methods employed in Long [5] and Bojnec and Ferto [6], which only explore the factors influencing the competitiveness of agricultural exports from an external perspective, the ERGM provides insights into the formation of relationships and the underlying factors that shape them, grounded in the nature and logic of network construction.

In contrast to traditional linear regression models, such as those presented by Zhou and Tong [4] and Huyen and Bang [10], ERGM does not exclude the issue of endogeneity. An essential category of explanatory variables in ERGM is network self-organization, which refers to the network structure of the explanatory variables. The underlying rationale is that network connections are both endogenous and interdependent [35].

We thank the reviewer for pointing to the interpretation of the regression results. 

In part of Basic regression analysis, we have made some addition about the interpretation of network effects:

This means that competitive export relationships are continually expanding. As a result, the 2-star of the export competitiveness network of provincial agricultural products contributes to the formation of network relationships. 

Provinces are likely to refrain from establishing numerous such relationships. Therefore, a negative 3-star coefficient indicates a “ceiling” on maintained friendships.

At the same time, an appropriate number of competitive export relationships is optimal.

In part of Basic regression analysis, we have made some deletion about the interpretation of network effects:

A positive 2-star coefficient suggests a propensity for forming new friendships, while a negative 3-star coefficient implies potential drawbacks from establishing and sustaining a large number of friendships.

In part of Basic regression analysis, we have made some addition about the interpretation of covariates:

These networks act as a valuable resource of social capital, providing natural advantages in enhancing bilateral trade, driving economic development and partnerships [67]. Specifically, by lowering language-related costs, sharing market information and consumer preferences, conducting agricultural cooperation projects, and optimizing supply chain management for agricultural products among sister cities, the sister-city networks ultimately build competitive advantages for agricultural exports. Consequently, playing the role of a sister-city platform has become a crucial strategy for many provinces and cities to promote their agricultural products.

4. Results and Discussion: The findings could be organized more logically and concisely. Avoid repetition and streamline the explanations to make them more accessible to readers. Discuss potential limitations of the study and suggestions for future research. For example, consider examining longer time series data or including additional covariates.

We thank the reviewer for pointing to this important result and discussion aspect of our study. In part of Robustness analysis, we have rephrased the sentences as follows:

The coefficients for pure structural effects and the lagged-by-one-period external covariate network are consistent with the benchmark regression results, and both pass the 1% significance test.

In part of Heterogeneity analysis, we have made some addition about the reasons for conducting heterogeneity analysis:

The influencing factors and effects may vary significantly among provinces due to differences in geographic location, resource endowment, economic development, and other factors.

Specific descriptions of regional divisions have been deleted in consideration of redundancy:

The eastern region includes provinces such as Beijing, Tianjin, Hebei, Shanghai, Jiangsu, Zhejiang, Fujian, Shandong, Guangdong, Hainan, Liaoning, Jilin, and Heilongjiang. The central region comprises Shanxi, Anhui, Jiangxi, Henan, Hubei, and Hunan, while the western region includes Inner Mongolia, Guangxi, Chongqing, Sichuan, Guizhou, Yunnan, Xizang, Shaanxi, Gansu, Qinghai, Ningxia and Xinjiang.

A sentence on regional divisions has been rephrased as follows:

This paper divides the 31 provinces into three regions-eastern, central, and western-based on their province location for the heterogeneity analysis.

In the Heterogeneity analysis section, the paragraph relating to Table 7 was changed as followed:

The results presented in Table 7 also reveal the heterogeneity in the impact of node attributes on the formation of competitive advantages in agricultural product exports across different regions. The CouGDP positively influences the competitiveness of agricultural product exports in the eastern provinces, but it has a negative effect in the central and western provinces. Agricultural exports from the eastern provinces benefit from lower transportation costs, making them more likely to be exported to developed markets, thereby facilitating their competitive edge. In contrast, the central and western provinces tend to export agricultural products to neighboring countries due to transportation cost constraints.

In the Heterogeneity analysis section, we have made some addition about the interpretation of node attribute indicators:

Since the western provinces must consider transportation costs, they prefer to export their competitive agricultural products to countries with open trade policies to minimize expenses.

In the Heterogeneity analysis section, the interpretation of international sister-city network has been rephrased as follows:

The international sister-city network shows a significant positive relationship in both categories. However, the coefficients for the international sister-city network are lower for countries along the Belt and Road. This result reveals that the friendly relations between Chinese provinces and countries along the Belt and Road are not entirely mirrored in the agricultural export competitiveness. Ultimately, the establishment of competitive agricultural export relationships by Chinese provinces along the Belt and Road is also shaped by contractual relationships, which diminish the influence of sister-city relations. Most countries along the Belt and Road possess natural geographical advantages and rely on foreign trade. Thus, there is a need for further examination into the role of international sister-city relationships in fostering competitive advantages in provincial agricultural exports to countries along the Belt and Road. Underpinned by a combination of national and local political relations, Chinese provinces find it easier to establish competitively advantageous agricultural export relationships.

As suggested, we addressed the deficiencies based on data and covariates.

Furthermore, the data in the China Customs database has been available since 2015, making the current timeframe under study relatively short. A longer span of data can enhance statistical credibility and improve the generalizability of the findings. Therefore, it is essential to consider examining longer time series data in future studies.

Moreover, while this paper incorporates three types of network structure statistics in the empirical regression analysis, some of these variables are not statistically significant. Future research will explore additional variables at the levels of Chinese provinces and countries to enhance its validity.

5. Policy Implications: The policy implications are well-aligned with the conclusions, but could be framed more concretely with specific actions provinces could take. Consider including real-world examples or case studies to illustrate the implementation of these policies.

We apologize for not elaborating policy implications concretely. As suggested, we have made some additions. 

For the first implications, the following phrases were added:

For example, Shandong Province, recognized as a significant agricultural export hub in China, can emphasize its strengths in exporting vegetables and fruits. Meanwhile, Zhejiang Province can stren

---

## [Decision Letter · Decision Letter 3]

4 Oct 2024

PONE-D-24-04543R3Export competitiveness network of Chinese provincial agricultural products: Evolution, performance, and influencing factorsPLOS ONE

Dear Dr. Chen,

Thank you for submitting your manuscript to PLOS ONE. After careful consideration, we feel that it has merit but does not fully meet PLOS ONE’s publication criteria as it currently stands. Therefore, we invite you to submit a revised version of the manuscript that addresses the points raised during the review process.

We look forward to receiving your revised manuscript.

Kind regards,

Tinggui Chen

Academic Editor

PLOS ONE

Journal Requirements:

Reviewers' comments:

Reviewer's Responses to Questions

**Comments to the Author**

1. If the authors have adequately addressed your comments raised in a previous round of review and you feel that this manuscript is now acceptable for publication, you may indicate that here to bypass the “Comments to the Author” section, enter your conflict of interest statement in the “Confidential to Editor” section, and submit your "Accept" recommendation.

Reviewer #4: All comments have been addressed

Reviewer #5: (No Response)

2. Is the manuscript technically sound, and do the data support the conclusions?

Reviewer #4: Yes

Reviewer #5: (No Response)

3. Has the statistical analysis been performed appropriately and rigorously? 

Reviewer #4: Yes

Reviewer #5: Yes

4. Have the authors made all data underlying the findings in their manuscript fully available?

Reviewer #4: Yes

Reviewer #5: Yes

5. Is the manuscript presented in an intelligible fashion and written in standard English?

Reviewer #4: Yes

Reviewer #5: Yes

6. Review Comments to the Author

Reviewer #4: （1）Improve the simplicity of language throughout the text.

（2）Please use standard English descriptions.

Reviewer #5: (No Response)

7. PLOS authors have the option to publish the peer review history of their article (what does this mean?). If published, this will include your full peer review and any attached files.

Reviewer #4: No

Reviewer #5: No

---

## [Author Response · Author response to Decision Letter 3]

14 Oct 2024

【We found that when pasting the changes here, the formatting of the changes could not be preserved. Therefore, the specific changes can be better presented in Response to Reviewers.docx.】

We are thankful to reviewer 4 for him/her valuable comments that helped to improve the manuscript.

Reviewer #4: 

 Improve the simplicity of language throughout the text.

We thank the reviewer for pointing to this important issue. We went through the whole text again and removed some redundant expressions.

A sentence in the third paragraph of the Introduction section was rephrased as follows:

These issues have impacted the competitiveness of Chinesea’s agricultural exports, revealing which have revealed a comparative disadvantage [3,4] that hinders both the export growth and the sustainable development of the sector [5].

A sentence in the fourth paragraph of the Introduction section was rephrased as follows:

Secondly, it enhances competitiveness in the global market and improves fosters the improvement of the international economic circulation system [7].

A sentence in the fifth paragraph of the Introduction section was rephrased as follows:

In light of these circumstances, tThe main objective of this study is to investigate the evolution, performance, status and influence influencing factors of the export competitiveness network of Chinese provincial export competitiveness for agricultural products through the perspective of complex networks. In this context, provinces refer to 27 Chinese provinces and 4 municipalities directly governed by the Central Government.

A sentence in the second paragraph of the External covariate effect section was rephrased as follows:

Furthermore, it reduces trade disputes and friction, thereby influencing the establishment and robustness of trade network relationships between cities and countries. However, bilateral geographical distance elevates the trade cost of trade, subsequently diminishing the establishment of competitively advantageous trading trade relationships. Based on these considerations, this study puts forth the following hypotheses:

We have combined the first paragraph of the Network construction section with the first paragraph of the Research design section. The first paragraph of the Research design section was rephrased as follows:

The article utilizes data from, using Chinese Customs data,to examines the dynamic evolution and competitive edge of agricultural exports across different Chinese provinces to various countries and regions. TheThis data was sourced from China Customs data, which has been available since 2015. By 2023, it may be possible to obtainobtaining complete data for the year ending in 2022 may be possible. Therefore, the research sample in this paper is based on data from 2015 to 2022 to ensure consistency and coherence.

Some sentences in the sixth paragraph of the Network construction section were rephrased as follows:

〖G(V,TCA)〗_t is an M*N matrix, where M representsis the composite nodes set of provinces and the agricultural products they export, and N denotesis the nodesset of destination countries. Therefore, the nodes of the export competitiveness network of Chinese provincial agricultural products include two categories: the composite nodes of agricultural products exported from Chinese provinces and the destination countries.

Since the first paragraph in the Construction of structural indicators section does not address the key elements, we have deleted this paragraph.

The export competitiveness network of provincial agricultural products is a typical two-mode network. This article primarily establishes the following structural indicators.

Some sentences in the first paragraph of the Network structure characteristics and evolution analysis section were rephrased as follows: 

Table 1 presents the structural indicators of the export competitiveness network of provincial agricultural products over time. The agricultural products exported from various provinces have proven to be competitive within expanding international markets, primarily evidenced by an increased in network edges and nodes. This suggests an overall improvement in the competitiveness of agricultural products.

A sentence in the fourth paragraph of the Network structure characteristics and evolution analysis section was rephrased as follows:

It is clear that The provinces selectively choose specific export markets for their agricultural products to cultivate a competitive advantage. However, this competitive advantage accounts for less than 4% across the 174 countries analyzed in this paper.

A sentence in the fifth paragraph of the Network structure characteristics and evolution analysis section was rephrased as follows:

Table 2 demonstrates a consistent The node degree among the top 5five countries in node degree perform consistently. 

A sentence in the paragraph before the Fig 3 was rephrased as follows:

However, the yearly rankings of the top 5 provincial agricultural products in degrees show variability and instability in competitive edge across provinces, except forwith only Guangdong Province consistently showing a competitive edge.

A sentence in the second paragraph of the Robustness analysis section was rephrased as follows:

Regression analyses of the export competitiveness network of Chinese provincial agricultural products from 2016 to 2022, based on a lagged-by-one-period external covariate network and node attribute indicators, demonstrate that the coefficients for pure structural effects and the lagged-by-one-period external covariate network coefficients closely align with the benchmark regression results,. The coefficients for pure structural effects and the lagged-by-one-period external covariate network are consistent with the benchmark regression results, and both pass the 1% significance test. Besides, the coefficients for Export also match the benchmark regression results. Hence, the regression utilizing the lagged-by-one-period external covariate network lagged term provides robust outcomes.

A sentence in the first paragraph before the Table 8 was deleted since the usefulness was not evident:

Most countries along the Belt and Road possess natural geographical advantages and rely on foreign trade.

Some sentences in the second paragraph of the Policy implications section were rephrased as follows:

(1) Enhancing and developing agricultural products with traditional advantages in for China's Chinese provinces, while strengthening collaboration with research institutes and relevant enterprises, which will cultivate distinctive and high-quality varieties, thereby and increaseing their international competitiveness and influenceof agricultural products.

Some sentences in the second paragraph of the Limitations section were rephrased as follows:

First, ERGM is primarily used to empirically analyze the influence of relationship establishment. However, future research should focus on empirically analyzing the impact of relationship strength , specifically exploring the factors that influence relationships strength withinon the export competitiveness network of Chinese provincial agricultural products.

（2）Please use standard English descriptions.

As suggested, we checked the full text and rephrased some expressions of the manuscript.

A sentence in the sixth paragraph of the Introduction section was rephrased as follows:

This paper, however, connects Chinese provinces with their exported agricultural products and examines the international competitiveness of Chinese provincial agricultural products at the provincial level, providing insights intofrom a more micro perspective to further explain the regionalization and differentiation characteristics of competitive agricultural products across Chinese provinces. 

Some sentences in the second paragraph of the Research on the two-mode network section were rephrased as follows:

Although some studies have explored the export competitiveness of Chinese agricultural products [5], none have assessed the influencing factors and evolutionevolutionary patterns of this competitiveness from a network perspective. This gap hinders our comprehensive understanding of the export networks of provincial agricultural products export networks. 

Based onUsing provincial export data, this articleit aims to establish a competitive network for agricultural product exports to various countries and regions. The objective is to analyze the evolution, performance, and influencing factors of the export competitiveness network of Chinese provincial agricultural productscompetitive advantages, challenges, and issues faced by agricultural product exports from different provinces within the international marketplace.

A sentence in the second paragraph of the Node attribute effect section was rephrased as follows:

Economic attributes significantly influence trade behavior and the overall structure of trade networks [39]

Some sentences in the second paragraph of the Network construction section were rephrased as follows:

This paper data helps builds a two-mode network of Chinese provincial agricultural product exports. The nodes in the two-modeexport network of Chinese provincial agricultural products comprises two types. The first type is a are composite nodes encompassing a provinces and a specific type of agricultural products, meaning that each node corresponds to a province and the agricultural product it exports.

A sentence in the third paragraph of the Number of network nodes, edges, density, and average degree section was rephrased as follows:

The closer the index is to 1, the more connected the network becomesAs the index approaches 1, the nodes within the network become denser, implying closer connections.

Some sentences in the third paragraph of the Network structure characteristics and evolution analysis section were rephrased as follows:

There is a notable potential for improving the competitive advantage of agricultural exports from Chinese provinces. While international trade networks usually exhibit high density [54], the density of this study constructed an the export competitiveness network o this study constructed f provincial agricultural products around a defined threshold. Consequently, the network’s density is remarkably low, consistently below 0.039, across the years studied. Therefore, boosting the competitiveness of these exports is imperative. Moreover, a consistent uptrend was noted in LPN, LPN(M) and LPN(N) the average degree across all classifications, indicateings an increase in the number of competitivedominant agricultural export partnerships initiated by Chinese provinces. On average, each province’s provincial agricultural exports product shows a competitive edge in merely five or six countries.

The first paragraph before Table 4 and the first paragraph after Table 4 were rephrased as follows:

This paper employed the stepwise method to perform an analysis aimed at The initial step involves identifying the most suitable regression model using the 2015 data and the specific regression results presented in . See Table 4. 

The star effects of the country were removed as the strong correlation when incorporating the star effects of both node types in the ERGM. AThe instability of the model results may be attributed to the strong correlation between the star effects of provincial agricultural products and the country in the ERGM. Consequently, it is not feasible to incorporate the pure structural effects of both node types into a single model. Simultaneously, as this paper primarily investigates the competitive advantage from the perspective of provincial agricultural products, the model only incorporates the star effect of provincial agricultural products.

A sentence in the second paragraph after the Table 7 was rephrased as follows:

The results presented in Table 7 also reveal the heterogeneity in the The impact of node attributes on formingthe formation of competitive advantages in agricultural product exports across different regions reveals the heterogeneity. 

A sentence in the sixth paragraph of the Policy implications section was rephrased as follows:

(4) Different regions should carry out reasonable regional layouts and planning for agricultural products to avoid homogenization and the proliferation of products that lack competitive strength.

---

## [Decision Letter · Decision Letter 4]

22 Oct 2024

PONE-D-24-04543R4Export competitiveness network of Chinese provincial agricultural products: Evolution, performance, and influencing factorsPLOS ONE

Dear Dr. Chen,

Thank you for submitting your manuscript to PLOS ONE. After careful consideration, we feel that it has merit but does not fully meet PLOS ONE’s publication criteria as it currently stands. Therefore, we invite you to submit a revised version of the manuscript that addresses the points raised during the review process.

We look forward to receiving your revised manuscript.

Kind regards,

Tinggui Chen

Academic Editor

PLOS ONE

Journal Requirements:

Reviewers' comments:

Reviewer's Responses to Questions

**Comments to the Author**

1. If the authors have adequately addressed your comments raised in a previous round of review and you feel that this manuscript is now acceptable for publication, you may indicate that here to bypass the “Comments to the Author” section, enter your conflict of interest statement in the “Confidential to Editor” section, and submit your "Accept" recommendation.

Reviewer #4: All comments have been addressed

Reviewer #5: (No Response)

2. Is the manuscript technically sound, and do the data support the conclusions?

Reviewer #4: Yes

Reviewer #5: Yes

3. Has the statistical analysis been performed appropriately and rigorously? 

Reviewer #4: Yes

Reviewer #5: Yes

4. Have the authors made all data underlying the findings in their manuscript fully available?

Reviewer #4: Yes

Reviewer #5: Yes

5. Is the manuscript presented in an intelligible fashion and written in standard English?

Reviewer #4: Yes

Reviewer #5: (No Response)

6. Review Comments to the Author

Reviewer #4: This paper presents a novel and valuable contribution to understanding the export competitiveness of Chinese agricultural products through a network lens. With revisions focused on clarity, methodological justification, and expansion of theoretical discussion, this study will be even more impactful.

1.Throughout the manuscript, there are several long and complex sentences, particularly in the introduction and methodology sections. Simplifying the language will enhance readability.

2.Clarity of H3 and H4: The hypotheses related to international sister-city networks and geographic distance (H3 and H4) are interesting, but these sections would benefit from additional theoretical background. Why are sister-city networks expected to play such a crucial role in agricultural product competitiveness? Adding a few supporting references would clarify the logic behind this.

3.While the Exponential Random Graph Model (ERGM) is appropriate for the study, the rationale behind choosing this model over other network models should be further elaborated. Highlighting its advantages in capturing interdependencies would reinforce its appropriateness.

4. The "star effects" and "node attribute effects" should be linked more explicitly to relevant studies in the literature review. This will ensure that the theoretical foundation for the hypotheses is clear.

5.While the policy implications section provides useful recommendations, it would benefit from being more specific. For example, how can regional governments leverage these findings to enhance their agricultural competitiveness? Additionally, consider linking the discussion to international trade policies or agreements.

Reviewer #5: (No Response)

7. PLOS authors have the option to publish the peer review history of their article (what does this mean?). If published, this will include your full peer review and any attached files.

Reviewer #4: No

Reviewer #5: No

---

## [Author Response · Author response to Decision Letter 4]

2 Dec 2024

We are thankful to reviewer 4 for him/her valuable comments that helped to improve the manuscript.

Reviewer #4: 

1.Throughout the manuscript, there are several long and complex sentences, particularly in the introduction and methodology sections. Simplifying the language will enhance readability.

We thank the reviewer for pointing to this important issue. We went through the whole text again and simplified some redundant sentences.

The second paragraph of the Introduction section was rephrased as follows:

In 2022, China’s Ministry of Agriculture and Rural Affairs rolled out the “14th Five-Year Plan for International Cooperation in Agriculture and Rural Affairs”, . This plan emphasizing emphasized the importance of strengthening international collaboration in agriculture and fostering competitive advantages in agricultural trade. Focusing on competitive advantage has now become a significant factor in China’s agricultural cooperation endeavors [1].

The third paragraph of the Introduction section was rephrased as follows:

Despite this progress, Chinese agriculture contends with challenges such as limited resources, land fragmentation, and high production costs [2]. These issues have led toimpacted the competitiveness a comparative disadvantage of in Chinese agricultural exports, revealing a comparative disadvantage [3,4] that and hinders hindered export growth and the sustainable development [5].

A sentence in the fifth paragraph of the Introduction section was rephrased as follows:

The main objective of this study is to investigate the evolution, performance, and influencing factors of the export competitiveness network of Chinese provincial agricultural products through the perspective of complex networks.

Some sentences in the sixth paragraph of the Introduction section have been rephrased as follows:

This paper, however, examines the international competitiveness of Chinese agricultural products at the provincial level, highlightingproviding insights into the regionalization and differentiation characteristics of competitive agricultural products across Chinese provinces.

This approach more effectively identifies competitive agricultural products with competitive advantages acrossin each provinces and further enriches the research paradigm and methodology concerningregarding the trade relationships.

Using the Exponential Random Graph Model (ERGM), this paper investigates the factors influencing the development of the export competitiveness network of provincial agricultural products from a comprehensive perspective. Compared to the conventionaltraditional linear regression models, ERGM is better suited to track the endogenous dependencies in forming the export competitiveness network, a task that conventional regression model models cannot effectively accomplish.

A sentence in the fourth paragraph of the Literature review section was rephrased as follows:

Furthermore, rising production costs in China and a diminishing demographic dividend, which the China country once benefited from, are undermining undermine its competitive edge in exporting agricultural products.

A sentence in the second paragraph of the External covariate effect section was rephrased as follows:

By enhancing knowledge of foreign markets [51], international sister-city relationships assist not only in grasping trade and investment opportunities but also in reinforcing identity and trust.

Some sentences in the second paragraph of the Research design section have been rephrased as follows:

This approach adopts topological indicators like loop numbers, node strength, clustering coefficient, and betweenness centrality expands upon social network analysis by to highlighting node heterogeneity, structural complexity, and dynamic variability in network patterns through topological indicators like loop numbers, node strength, average clustering coefficient, and betweenness centrality.

Therefore, employing complex network analysis can effectively illuminate the structural evolution of the export competitiveness network of Chinese provincial agricultural products and identify the competitive agricultural products.

In contrast to traditional research methods, complex network analysis can investigate the key factors influencing agricultural export competitiveness at various levels. Based on this, we can provide, ultimately providing actionable recommendations to enhance the export competitiveness of these products in Chinese provinces.

A sentence in the fifth paragraph of the Network construction section was rephrased as follows:

Instead, it focuses on the share of specific agricultural exports in the province and the share of national imports of specific agricultural products in total imports.competitiveness of a particular agricultural product exported by the province in comparison to all agricultural product exported by that province and those of all other provinces.

The paragraph about Node attribute indicators in Empirical analysis was rephrased as follows:

Node attribute indicators. At the provincial level, this paper accounts for GDP, the total output value of agriculture, forestry, animal husbandry, and fishery (AFAHF), and exports [71]. Each province's The GDP and export data for each province are sourced from the China Statistical Yearbook, while the AFAHF data come from the China Rural Statistical Yearbook. Nationally, it is important to account for GDP, population [72], and economic freedom [73] attributes. Each country'sThe GDP and population data for each country are derived from the World Bank’s World Development Indicators database, while data about economic freedom is provided by the Global Heritage Foundation provides data about economic freedom.

A sentence in the third paragraph of the Analysis of regression results section was rephrased as follows:

China has emphasized the importance of promoting high-quality agricultural development, paying more attention to expanding favorable agricultural products in international markets, and creating new competitivenesscompetitive advantages in for exporting specialty agricultural products in international markets.

Some sentences in the second paragraph of the Policy implications section have been rephrased as follows:

Enhancing and developing agricultural products with traditional advantages for in Chinese provinces while strengthening collaboration with research institutes and relevant enterprises, . which will cultivate distinctive and high-quality varieties and increase the international competitiveness of agricultural products.

For example, Shandong Province, recognized as a significant agricultural export hub in China, can emphasize its strengths in exporting vegetables and fruits. Meanwhile, Zhejiang Province can strengthen the international competitiveness of its tea industry, and Yunnan Province can focus on building its strengths in flower exports, among other initiatives.

Some sentences in the last two paragraphs of the Policy implications section have been rephrased as follows:

It is essential to leverage international sister-city relationships to create new opportunities for bilateral economic and trade cooperation and to enhance the competitivenesscompetitive advantage of provincial agricultural exports in friendly countries. International agricultural cooperation agreements Interactions and collaborations between provinces and global partners should be encouraged to increase awareness and recognition of provincial agricultural products through bilateral cultural and economic exchanges.

Different regions should carry out reasonable regional planning for agricultural products to avoid competition of homogenized agricultural products in the same markethomogenization and the proliferation of products that lack competitive strength. The eastern region should fully utilize its geographic advantages by targetingto consolidate its dominant position in agricultural markets such as the United States, the European Union, Japan, and South Korea, and other countries as the main export destinations for its advantageous agricultural products. Meanwhile, the central and western regions should actively promote the diversification of agricultural export markets based on the China Railway Express and the new western land-sea corridor. By differentiating the layout of agricultural products in different markets, it is easier to form a competitive advantage of agricultural products.

2.Clarity of H3 and H4: The hypotheses related to international sister-city networks and geographic distance (H3 and H4) are interesting, but these sections would benefit from additional theoretical background. Why are sister-city networks expected to play such a crucial role in agricultural product competitiveness? Adding a few supporting references would clarify the logic behind this.

We thank the reviewer for pointing to this important theoretical aspect of our study. In External covariate effect section, we have added some references to support H3 and H4. In the first paragraph:

Individuals can leverage social capital relationships to enhance their advantages and seize opportunities [49].

In the second paragraph:

Sister-city relationships are an advantageous resource [50].

The quality of information and resources transmitted by such strong relationships generated by identification and trust tends to be higher, providing both parties with a competitive advantage in trade and economic activities [43]. 

Furthermore, it reduces trade disputes and friction, thereby influencing the establishment and robustness of trade relationships between cities and countries [52]. As a result, products from friendly cities or provinces have a significant advantage in entering the markets of friendly countries.

In the third paragraph:

Geographic location influences the development of relationships [22]. Usually, distances are used to proxy costs at an abstract level [53]. As a result, coastal cities possess a greater geographical advantage when exporting their products [54]. Longer However, bilateral geographical distances increaseelevates the trade costs, thereby becoming an impediment to product exports [55] and further hindering diminishing the establishment of competitively advantageous trade relationships.

3.While the Exponential Random Graph Model (ERGM) is appropriate for the study, the rationale behind choosing this model over other network models should be further elaborated. Highlighting its advantages in capturing interdependencies would reinforce its appropriateness.

The reasons for using ERGM instead of other models have been elaborated upon in the Introduction and Construction of ERGM and variable explanation sections. As proposed by the reviewer, we provide additional explanations.

In the third and fourth paragraphs of Construction of ERGM and variable explanation section, we have made some additions.

It is indeed true that established product export relationships will further accumulate experience for the product's overseas expansion and facilitate its export to additional countries. Newly established export relationships are somewhat dependent on existing ones.

Relative to other network models, such as the Quadratic Assignment Procedure (QAP), which focuses on the correlation and regression between matrices [65], this model does not take into account for the effects of node attributes and covariate networks on relationship development. Similarly, the Simulation Investigation for Empirical Network Analysis (SIENA) primarily incorporates explanatory variables related to structural effects and individual actor attribute effects [66]. However, it also fails to consider external covariate networks.

4. The "star effects" and "node attribute effects" should be linked more explicitly to relevant studies in the literature review. This will ensure that the theoretical foundation for the hypotheses is clear.

As suggested, in the Research on the network of agricultural product export section, we have incorporated new literature related to star effects and node attribute effects.

Countries with high betweenness centrality serve as hubs in agricultural trade networks and tend to establish a greater number of trade relationships [20]. This phenomenon results in more network star effects [21]. Furthermore, the star effect promotes the establishment of additional network relationships [22]. In addition, attribute variables such as factor endowment, economic size, and geographic location are critical factors influencing the expansion of agricultural trade networks [23].

5.While the policy implications section provides useful recommendations, it would benefit from being more specific. For example, how can regional governments leverage these findings to enhance their agricultural competitiveness? Additionally, consider linking the discussion to international trade policies or agreements.

We thank the reviewer for raising this important point. In the second paragraph of Policy implications section, a sentence has been rephrased as follows:

Specifically, provincial governments can improve the quality of agricultural products through standardized, large-scale, and modernized production models, thereby further strengthening the advantages of traditional agricultural products. 

We have made some additions in the second paragraph:

Specifically, the government can increase funding and concentrate on fostering scientific and technological innovation within agribusinesses. It is essential to establish a high-quality team of experts in the field of agriculture, consisting of faculty and students from universities, as well as agricultural scientists. Furthermore, emphasis should be placed on fostering collaboration among local governments, agricultural universities, and agriculture-related institutions and enterprises.

Some sentences have been rephrased as follows in the fourth paragraph of Policy implications section:

On the other hand, local governments should leverage major agribusinesses as the foundation for agricultural industrialization to achieve economies of scale.provincial governments can enhance the export of advantageous agricultural products by building stable and mature export routes.

---

## [Decision Letter · Decision Letter 5]

6 Dec 2024

Export competitiveness network of Chinese provincial agricultural products: Evolution, performance, and influencing factors

PONE-D-24-04543R5

Dear Dr. Chen,

We’re pleased to inform you that your manuscript has been judged scientifically suitable for publication and will be formally accepted for publication once it meets all outstanding technical requirements.

Kind regards,

Tinggui Chen

Academic Editor

PLOS ONE

Additional Editor Comments (optional):

Reviewers' comments:

Reviewer's Responses to Questions

**Comments to the Author**

1. If the authors have adequately addressed your comments raised in a previous round of review and you feel that this manuscript is now acceptable for publication, you may indicate that here to bypass the “Comments to the Author” section, enter your conflict of interest statement in the “Confidential to Editor” section, and submit your "Accept" recommendation.

Reviewer #5: All comments have been addressed

2. Is the manuscript technically sound, and do the data support the conclusions?

Reviewer #5: Yes

3. Has the statistical analysis been performed appropriately and rigorously? 

Reviewer #5: Yes

4. Have the authors made all data underlying the findings in their manuscript fully available?

Reviewer #5: Yes

5. Is the manuscript presented in an intelligible fashion and written in standard English?

Reviewer #5: (No Response)

6. Review Comments to the Author

Reviewer #5: (No Response)

7. PLOS authors have the option to publish the peer review history of their article (what does this mean?). If published, this will include your full peer review and any attached files.

Reviewer #5: No

---

## [Editor Report · Acceptance letter]

8 Jan 2025

PONE-D-24-04543R5 

PLOS ONE

Dear Dr. Chen, 

I'm pleased to inform you that your manuscript has been deemed suitable for publication in PLOS ONE. Congratulations! Your manuscript is now being handed over to our production team.

Kind regards, 

on behalf of

Dr. Tinggui Chen 

Academic Editor

PLOS ONE